# LANGUAGE-MEDIATED, OBJECT-CENTRIC REPRESENTATION LEARNING

## ABSTRACT

We present Language-mediated, Object-centric Representation Learning (LORL), a paradigm for learning disentangled, object-centric scene representations from vision and language. LORL builds upon recent advances in unsupervised object segmentation, notably MONet and Slot Attention. While these algorithms learn an object-centric representation just by reconstructing the input image, LORL enables them to further learn to associate the learned representations to concepts, i.e., words for object categories, properties, and spatial relationships, from language input. These object-centric concepts derived from language facilitate the learning of object-centric representations. LORL can be integrated with various unsupervised segmentation algorithms that are language-agnostic. Experiments show that the integration of LORL consistently improves the object segmentation performance of MONet and Slot Attention on two datasets via the help of language. We also show that concepts learned by LORL, in conjunction with segmentation algorithms such as MONet, aid downstream tasks such as referring expression comprehension.

## 1 INTRODUCTION

Cognitive studies show that human infants develop object individuation skill from diverse sources of information: spatial-temporal information, object property information, and language (Xu, 1999; 2007; Westermann & Mareschal, 2014). Specifically, young infants develop object-based attention that disentangles the motion and location of objects from their visual appearance features. Later on, they can leverage the knowledge acquired through word learning to solve the problem of object individuation: words provide clues about object identity and type. The general picture from cognitive science is that object perception and language co-develop in support of one another (Bloom, 2002).

Our long-term goal is to endow machines with similar abilities. In this paper, we focus on how language may support object segmentation. Many recent works have studied the problem of unsupervised object representation learning, though without language. As an example, factorized, object-centric scene representations have been used in various kinds of prediction (Goel et al., 2018), reasoning (Yi et al., 2018; Mao et al., 2019), and planning tasks (Veerapaneni et al., 2020), but they have not considered the role of language and how it may help object representation learning.

As a concrete example, consider the input images shown in Fig. 1 and the paired questions. From language, we can learn to associate concepts, such as *black, pan*, and *legs*, with the referred object's visual appearance. Further, language provides cues about how an input scene should be segmented into individual objects: a wrong parsing of the input scene will lead to an incorrect answer to the question. We can learn from such failure that the handle belongs to the frying pan (Fig. 1a) and the chair has four legs (Fig. 1b).

Motivated by these observations, we propose a computational learning paradigm, Language-mediated, Object-centric Representation Learning (LORL), associating learned object-centric representations to their visual appearance (masks) in images, and to concepts—words for object properties such as color, shape, and material—as provided in language. Here the language input can be either descriptive sentences or question-answer pairs. LORL requires no annotations on object masks, categories, or properties during the learning process.

In LORL, four modules are jointly trained. The first is an image encoder, learning to encode an image into factorized, object-centric representations. The second is an image decoder, learning to

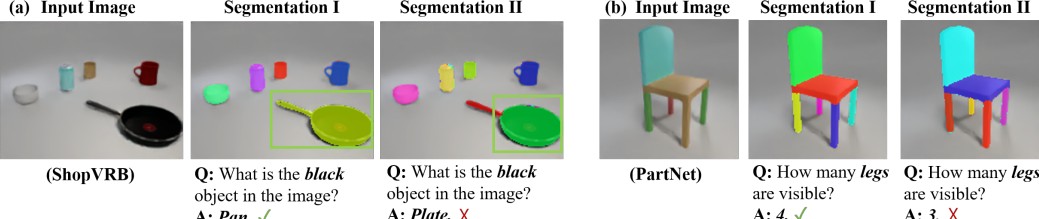

Figure 1: Two illustrative cases of Language-mediated, Object-centric Representation Learning. Different colors in the segmentation masks indicate individual objects recognized by the model. LORL can learn from visual and language inputs to associate various concepts: *black*, *pan*, *leg* with the visual appearance of individual objects. Furthermore, language provides cues about how an input scene should be segmented into individual objects: (a) segmenting the frying pan and its handle into two parts yields an incorrect answer to the question (Segmentation II); (b) an incorrect parsing of the chair image makes the counting result wrong (Segmentation II).

reconstruct masks for individual objects from the learned representations by reconstructing the input. These two modules share the same formulation as recent unsupervised object segmentation research, and may be realized as methods such as MONet (Burgess et al., 2019) and Slot Attention (Locatello et al., 2020).

The third module in LORL is a pre-trained semantic parser that translates the input sentence into a semantic, executable program, where each concept (i.e., words for object properties such as 'red') is associated with a vector space embedding. Finally, the last module, a neural-symbolic program executor, takes the object-centric representation from Module 1, intermediate representations from Module 2, and concept embeddings and the semantic program from Module 3 as input, and outputs an answer if the language input is a question, or TRUE/FALSE if it's a descriptive sentence. The correctness of the executor's output and the quality of reconstructed images (as output of Module 2) are the two supervisory signals we use to jointly train Modules 1, 2, and 4.

We integrate the proposed LORL with state-of-the-art unsupervised segmentation methods, MONet (Burgess et al., 2019) and Slot Attention (Locatello et al., 2020). The evaluation is based on two datasets: ShopVRB (Nazarczuk & Mikolajczyk, 2020) contains images of daily objects and question-answer pairs; PartNet (Mo et al., 2019) contains images of furniture with hierarchical structure, supplemented by descriptive sentences we collected ourselves. We show that LORL consistently improves existing methods on unsupervised object segmentation, much more likely to group different parts of a single object into a single mask.

We further analyze the object-centric representations learned by LORL. In LORL, conceptually similar objects (e.g. objects of similar shapes) appear to be clustered in the embedding space. Moreover, experiments demonstrate that the learned concepts can be used in new tasks, such as visual grounding of referring expressions, without any additional fine-tuning.

## 2 RELATED WORK

**Unsupervised object representation learning.** Given an input image, unsupervised object representation learning methods segment objects in the scene and build an object-centric representation for them. A mainstream approach has focused on using compositional generative scene models that decompose the scene as a mixture of component images (Greff et al., 2016; Eslami et al., 2016; Greff et al., 2017; Burgess et al., 2019; Engelcke et al., 2020; Greff et al., 2019; Locatello et al., 2020). In general, these models use an encoder-decoder architecture: the image encoder encodes the input image into a set of latent object representations, which are fed into the image decoder to reconstruct the image. Specifically, Greff et al. (2019); Burgess et al. (2019); Engelcke et al. (2020) use recurrent encoders that iteratively localize and encode objects in the scene. Another line of research (Eslami et al., 2016; Crawford & Pineau, 2019; Kosiorek et al., 2018; Stelzner et al., 2019; Lin et al., 2020) leverages object locality to attend to different local patches of the image. These models often use a pixel-level reconstruction loss. In contrast, we propose to explore how language, in addition to visual observations, may contribute to object-centric representation learning. There have also been work that uses other types of supervisions, such as dynamic prediction (Kipf et al., 2020; Bear et al.,

2020) and multi-view consistency (Prabhudesai et al., 2020). In this paper, we focus on unsupervised learning of object-centric representations from static images and language.

**Visual concept learning.** Learning visual concepts from language and other forms of supervisions provides useful representations for various downstream tasks, such as cross-modal retrieval (Wu et al., 2019), visual-question answering (Yi et al., 2018; Mao et al., 2019), and scene manipulation (Prabhudesai et al., 2020). Previous work has been focusing on various types of representations (Ren et al., 2016; Wu et al., 2017), training algorithms (Faghri et al., 2018) and supervisions (Johnson et al., 2016; Yang et al., 2018). In this paper, we focus on learning visual concepts that can be grounded on object-centric representations. Recent works on object-centric grounding of visual concepts (Wu et al., 2017; Mao et al., 2019; Prabhudesai et al., 2020) have shown great success in achieving high performance in downstream tasks and strong generalization from a small amount of data. However, these methods assume pre-trained object detectors to generate object proposals in the scene. In contrast, our LORL learns to individuate objects and associates concepts with the learned object-centric representations without any annotations on object segmentation masks or properties.

## 3 PRELIMINARIES

Before delving into our language-mediated object-centric representation learning paradigm, we first discuss a general formulation that unifies multiple concurrent unsupervised object representation learning methods and a neuro-symbolic framework for learning visual concepts from language.

### 3.1 UNSUPERVISED OBJECT-CENTRIC REPRESENTATION LEARNING

Given an image $I$, a typical unsupervised object representation learning model will decompose the scene into a series of *slot* profiles $\{(z_1, x_1, m_1), \ldots, (z_K, x_K, m_K)\}$, where each slot profile is expected to represent an object (or nothing, as the number of slots may be greater than the actual number of objects in the scene). Here $z_i$ is the object feature, $x_i$ is the object image, and $m_i$ is the object mask specifying its location in the scene.

In our paper, we focus on two recent models, MONet (Burgess et al., 2019) and Slot Attention (Locatello et al., 2020). MONet uses a recurrent spatial attention network (Ronneberger et al., 2015) to segment out objects in the scene, and adopts a variational autoencoder (Kingma & Welling, 2014) to encode objects as well as reconstructing object images for self-supervision. At a very high level, its objective function is calculated as

$$\mathcal{L} = \left\| \left( \sum_{k=1}^{K} m_k x_k \right) - I \right\|_2^2 + \beta \cdot \sum_{k=1}^{K} \mathrm{KL}(z_k),$$ (1)

where the first term is a pixel-wise L2 reconstruction loss and the second term computes the KL divergence between the distribution of $z_k$'s and a prior Gaussian distribution.

Slot Attention uses a transformer-like attention network (Vaswani et al., 2017) to extract object features, and decode them with convolutional neural networks to component images and object masks. The model is trained by the same reconstruction loss in the form of L2-norm:

$$\mathcal{L} = \left\| \left( \sum_{k=1}^{K} m_k x_k \right) - I \right\|_2^2.$$ (2)

### 3.2 NEURO-SYMBOLIC CONCEPT LEARNING

The neuro-symbolic concept learner (NS-CL; Mao et al., 2019) learns visual concepts by looking at images and reading paired questions and answers. NS-CL takes a set of segmented objects in a given image as its input, extracts their visual features with a ResNet (He et al., 2015), translates the input question into an executable program by a semantic parser, and executes the program based on the object-centric representation to answer the question. The key idea of NS-CL is to explicitly represent individual concepts in natural language (colors, shapes, spatial relationships, etc.) as vector space embeddings, and associate them with the object embeddings.

NS-CL answers the input question by executing the program based on the object-centric representation. For example, in order to query the name of the white object in Fig. 2, NS-CL first filters out the object by computing the cosine similarity between the concept *white* and individual object representations, which produces a "mask" vector where each entry denotes the probability that an object has been

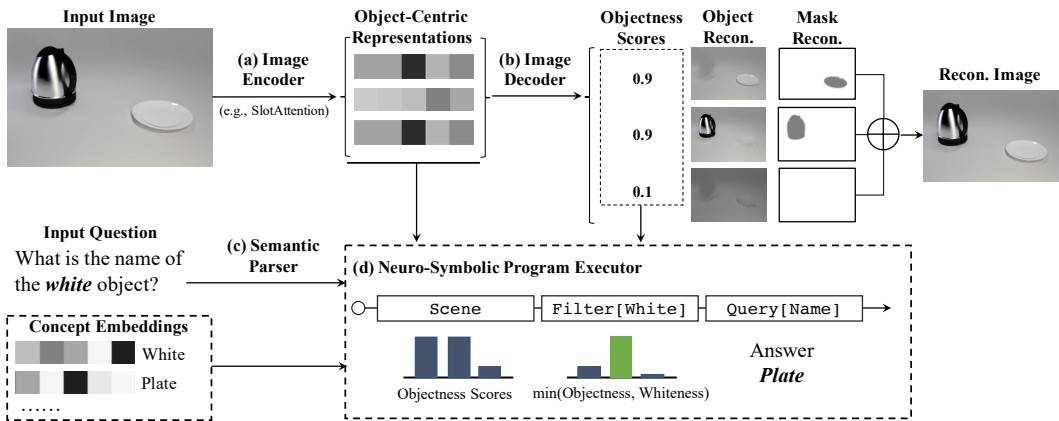

Figure 2: Our LORL contains four modules. (a) An image encoder encodes the input image into a factorized, object-centric representation. (b) An image decoder learns to reconstruct the image from the learned representations. It also decodes an objectness score based on the representation. (c) A pre-trained semantic parser translates the input sentence into an executable program and associates concepts with learnable vector embeddings. (d) A neuro-symbolic program executor takes the object-centric representation, the objectness scores, the parsed program, and concept embeddings as input to predict the answer (if the input is a question) or `TRUE`/`FALSE` (if the input is a descriptive sentence).

selected. The output "mask" on the objects will be fed into the next module and the execution will continue. The last query operation will produce the answer to the question. The vector embeddings of individual objects and the concepts will be jointly trained based on language supervision.

## 4 Language-mediated, Object-centric Representation Learning

Marrying the ideas of unsupervised object-centric representation learning and neuro-symbolic concept learning, we are able to learn an object-centric representation using both visual and language supervision. Fig. 2 shows an overview of Language-mediated, Object-centric Representation Learning (LORL). In LORL, four modules are optimized jointly: an image encoder, an image decoder, a semantic parser, and a neuro-symbolic program executor.

**Image encoder.** Given an input image, we first use the image encoder (Fig. 2a) to individuate objects in the scene and extract an object-centric scene representation. It takes the input image as its input, individuates objects in the scene, and produces a collection of latent slot embeddings $\{z_i\}$.

**Image decoder.** The decoder (Fig. 2b) takes the object-centric representation produced by the image encoder and produces a 3-tuple for each individual slot $(x_k, m_k, s_k)$, where $x_k$ reconstructs the RGB image of the slot, $m_k$ reconstructs the mask, and $s_k \in [0, 1]$ is a scalar indicating the *objectness* of the slot. That is, whether $k$-th slot corresponds to a *single* object in the scene. Here, we have extended the general pipeline we described in Section 3.1 with an objectness indicator. It serves dual purposes. First, it weights each reconstructed component image while generating the reconstructed image. Mathematically, the reconstructed image $I'$ is computed as: $I' = \sum_{k=1}^{K} s_k \cdot (m_k x_k)$. Second, it mediates the output of all `filter` operations in the program executor.

In this paper, we will experiment with two image encoder-decoder options: MONet (Burgess et al., 2019) and Slot Attention (Locatello et al., 2020). They are both compatible with the learning paradigm described above. For both models, we use a single linear layer to predict the objectness score for each slot on top of the second-last layer of their image decoders.

**Semantic parser.** A pre-trained semantic parser (Fig. 2c) will translate the input question into an executable program composed of primitive operations, such as `filter`, which filters out objects with certain concepts and `query`, which queries the attribute of the input object. We use roughly the same domain-specific language (DSL) for representing programs as CLEVR (Johnson et al., 2017a, see also Appendix A for details). All concepts that appear in the program, such as *white*, are associated with distinct, learnable concept embedding vectors.

**Neuro-symbolic program executor.** The program executor (Fig. 2d) takes the object-centric representation from the image encoder $\{z_k\}$, the objectness score $\{s_k\}$ from the image decoder, the concept embeddings and the program generated by the semantic parser as its input. It executes the program based on the visual and concept representations to answer the question.

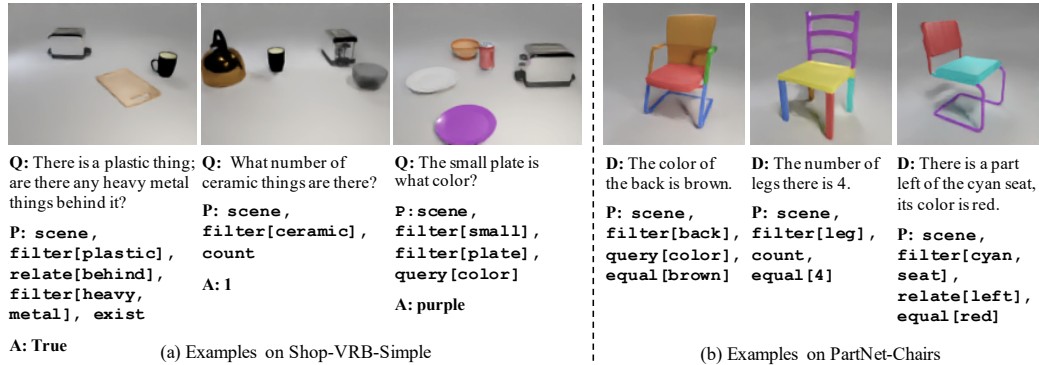

**Q:** There is a plastic thing; are there any heavy metal things behind it?

**P: scene, filter[plastic], relate[behind], filter[heavy, metal], exist**

**A: True**

**Q:** What number of ceramic things are there?

**P: scene, filter[ceramic], count**

**A: 1**

**Q:** The small plate is what color?

**P:scene, filter[small], filter[plate], query[color]**

**A: purple**

(a) Examples on Shop-VRB-Simple

**D:** The color of the back is brown.

**P: scene, filter[back], query[color], equal[brown]**

**D:** The number of legs there is 4.

**P: scene, filter[leg], count, equal[4]**

**D:** There is a part left of the cyan seat, its color is red.

**P: scene, filter[cyan, seat], relate[left], equal[red]**

(b) Examples on PartNet-Chairs

Figure 3: In Shop-VRB-Simple, questions involve seven different attributes of objects in the scene: *name, size, weight, material, color, shape, mobility*. In PartNet-Chairs, images are paired with sentences describing the *names* and *colors* of the parts in the scene, and their relationships.

The original program executor in NS-CL (Section 3.2) assumes a pre-trained object detector for generating object proposals. In LORL, we associate each object representation with an objectness score $s_k$. Recall that a `filter` operation in NS-CL produces a mask vector indicating whether an object has been selected. Here, we mediate the output of an `filter(c)` operation as $\min(s_k, \text{filter(c)})$. Intuitively, a slot will be selected only if, first, it has concept $c$ and, second, it corresponds to a single object in the scene.

**Training paradigm.** During training, we jointly optimize the image encoder, the image decoder, and the concept embeddings. They are trained by minimizing the loss $\mathcal{L}$:

$$\mathcal{L} = \alpha \cdot \mathcal{L}_{\text{perception}} + \beta \cdot \mathcal{L}_{\text{reasoning}}.$$

For MONet-based image encoder-decoder, we use Equation 1 as the perception loss $\mathcal{L}_{\text{perception}}$, while for Slot Attention-based encoder-decoder, we use Equation 2. The neuro-symbolic program executor produces a distribution over candidate answers to the input question. We use the cross-entropy loss between the predicted answer and the ground truth answer as $\mathcal{L}_{\text{reasoning}}$.

We use a three-stage training paradigm in LORL. First, we train the model with only visual inputs with $\mathcal{L}_{\text{perception}}$ for $N_1$ epochs. Next, we fix the image encoder and the image decoder, and optimize the concept embeddings with the loss term $\mathcal{L}_{\text{reasoning}}$ for $N_2$ epochs. During this second stage, the image encoder and the decoder can already produce descent object segmentation results. Finally, we jointly optimize all three modules for $N_3$ epochs. We provide detailed information about the hyperparameters for different models in Appendix B.

## 5 EXPERIMENTS

We first evaluate whether the representations learned by LORL lead to better image segmentation with the help of language. We then evaluate how these representations may be used for instance retrieval, visual reasoning, and referring expression comprehension.

### 5.1 IMAGE SEGMENTATION

**Data.** We use two datasets for image segmentation evaluation. The first, Shop-VRB-Simple, is based on Shop-VRB (Nazarczuk & Mikolajczyk, 2020), a dataset of complex household objects and question-answer pairs. The second is based on chairs in PartNet (Mo et al., 2019), a dataset where the objects are different parts of a chair. Fig. 3 shows some examples from the two datasets.

Shop-VRB is a visual reasoning dataset, similar to CLEVR (Johnson et al., 2017a), but with complex household objects of different sizes, weights, materials, colors, shapes, and mobility. Because the original Shop-VRB dataset includes very small and highly transparent objects and complex backgrounds, which current unsupervised representation learning models cannot handle, we generate 10K images with a clean background ourselves using large objects from the dataset. We also pair every image with 9 questions, resulting in 90K questions in total. The test split has 960 images and 8.6K questions. We name this variant Shop-VRB-Simple.

| (a) | ARI ↑ | GT Split ↓ | Pred Split ↓ |
|---|---|---|---|
| Slot Attention (SA) | 83.51±2.3 | 15.68±1.9 | 13.19±1.5 |
| LORL + SA | **89.23±1.6** | **9.95±1.6** | **10.18±1.3** |

| (b) | Slot Attn. | LORL + SA |
|---|---|---|
| Coffee maker | $39.4_{\pm 7.0}$ | $\mathbf{21.9_{\pm 8.3}}$ |
| Blender | $38.9_{\pm 11.5}$ | $\mathbf{17.7_{\pm 3.4}}$ |
| Toaster | $33.4_{\pm 3.9}$ | $\mathbf{16.4_{\pm 7.2}}$ |

Table 1: (a) Results on Shop-VRB-Simple. LORL helps to improve Slot Attention Locatello et al. (2020) in all metrics. (b) The Ground Truth split ratio for the three object categories where Slot Attention most commonly fail. LORL helps Slot Attention to reduce the ratio by 50%.

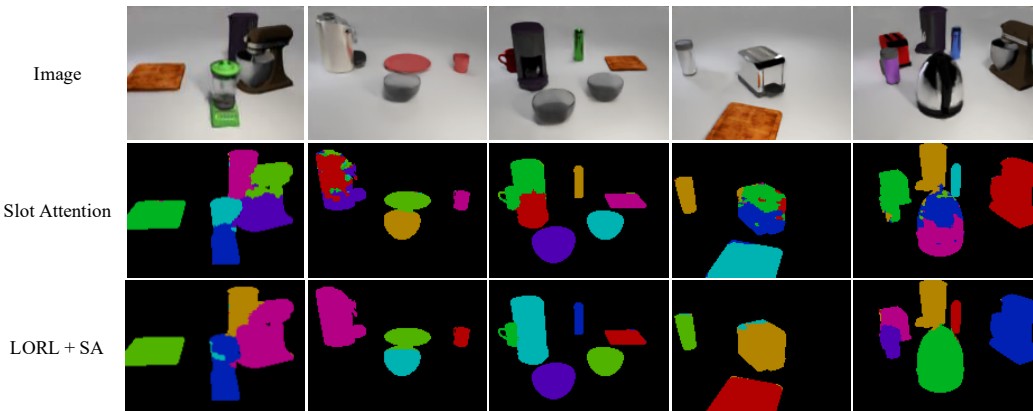

Figure 4: Visualization on Shop-VRB-Simple. Pixels with the same color represent a mask produced by the models. The Slot Attention model often fails to segment blenders, coffee makers, and toasters. LORL helps to greatly improve its results.

While the previous literature on unsupervised object segmentation mainly focuses on settings where objects are spatially disentangled, we also explore how language may help when objects of interest are different parts of a global shape. To this end, we collect a new dataset, PartNet-Chairs, using chair shapes from PartNet. Every image here shows a chair, where each part of the chair (legs, seat, back, arms) is randomly assigned a color. We select six different chair shapes with one or four legs and zero or two arms. We generate 5K images for training. Each image is paired with 8 descriptive sentences generated from human-written templates, resulting in 40K examples in total. The test split has 960 images. Each sentence describes the *name* and *color* of parts. We provide all templates in Appendix C. We are interested in whether object-centric representation learning models may separate these parts and whether and how language may help in this scenario.

**Baselines.** We use MONet (Burgess et al., 2019) and Slot Attention (Locatello et al., 2020) as the image modules (Modules 1 and 2), and evaluate how the incorporation of language may improve their performance. Because MONet is color-sensitive, and on Shop-VRB-Simple, many objects have diverse colors and specular reflection, it does not produce meaningful results there. Thus we only show results with MONet on PartNet-Chairs. We show results with Slot Attention on both datasets.

**Metrics.** We use three metrics for evaluation. Following Greff et al. (2019), we first use the Adjusted Rand Index (ARI; Rand, 1971; Hubert & Arabie, 1985). It treats segmentation as a clustering problem: each mask is the cluster index that the pixels within belong to. ARI is computed as the similarity between the predicted and ground truth clusters, and ranges from 0 (random) to 1 (perfect match).

In practice, we found this pixel-wise metric is sensitive to the size of objects: a model that infrequently makes mistakes on large objects will have lower ARI than one that frequently mis-segments small objects. Thus, we in addition design two object-centric metrics:

- **Ground Truth Split Ratio** (GT Split) measures the ratio of objects (GT masks) that are covered by more than one prediction mask.
- **Prediction Split Ratio** (Pred Split) measures the ratio of prediction masks that cover more than one object (GT mask).

Concretely, we first assign each pixel to the prediction mask with the maximum value at the pixel. We say a prediction mask covers an object if it covers at least 20% of the object's pixels. The GT and Pred Split ratios are thus defined as:

|  | ARI ↑ | GT Split ↓ | Pred Split ↓ |
|---|---|---|---|
| MONet | $91.41_{\pm 3.7}$ | $10.3_{\pm 5.1}$ | $14.09_{\pm 5.2}$ |
| LORL + MONet | $94.91_{\pm 2.1}$ | $4.95_{\pm 0.7}$ | $4.02_{\pm 2.5}$ |
| Slot Attention | $87.32_{\pm 3.6}$ | $12.54_{\pm 6.6}$ | $22.99_{\pm 5.0}$ |
| LORL + Slot Attention | $95.81_{\pm 1.0}$ | $3.39_{\pm 1.1}$ | $2.92_{\pm 1.0}$ |

Table 2: Quantitative results on PartNet-Chairs. All numbers are in percentage. LORL consistently improves MONet's and Slot Attention's performance on segmentation.

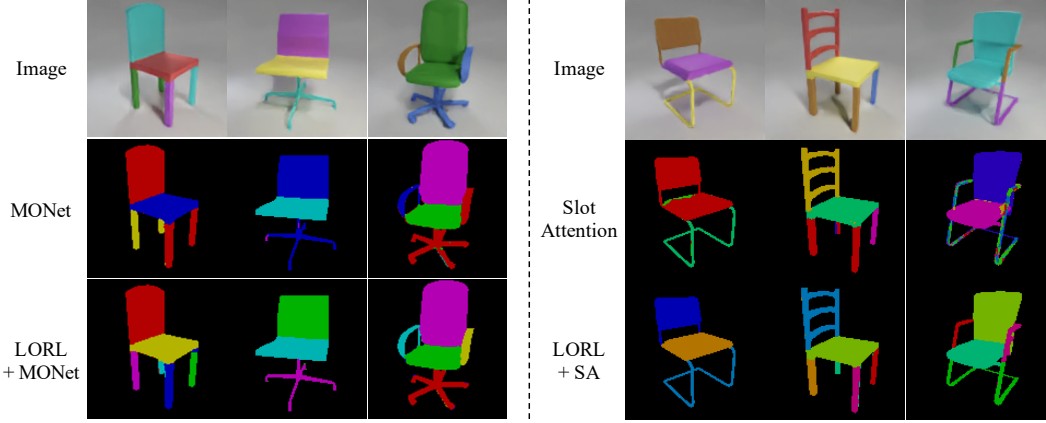

Figure 5: Visualization on PartNet-Chairs. Pixels with the same color represent a mask produced by the models. LORL successfully recognizes different parts in various situations.

$$\text{GTSplit} = \frac{\text{\# of objects that are covered by} > 1 \text{ masks}}{\text{\# of objects that are covered by} > 0 \text{ masks}}; \ \text{PredSplit} = \frac{\text{\# of masks that cover} > 1 \text{ objects}}{\text{\# of masks that cover} > 0 \text{ objects}}.$$

Ideally, there is a one-to-one correspondence between objects and predicted masks, with both GT Split and Pred Split being 0.

**Results.** The quantitative results on SHOP-VRB-Simple are summarized in Table 1. We show the mean and standard error on each metric over 3 runs. Since our semantic parsing module is trained on paired question-program pairs, it achieves nearly perfect accuracy ($> 99.9\%$) on test questions. Thus, in later sections, we will focus on evaluating object segmentation, concept grounding, and downstream task performances. LORL helps Slot Attention achieve better segmentation results in all three metrics. From visualizations in Fig. 4, we find that the original Slot Attention model struggles with metallic objects; but with LORL, it performs much better in those cases.

To further explore how LORL helps Slot Attention on failure cases, we calculate the Ground Truth Split Ratio for each object category, and find that Slot Attention most often fail to segment coffee makers, blenders, and toasters as a whole. These objects have complex sub-parts and their appearance changes quickly when the viewpoint changes. With the help of language, Slot Attention improves consistently over its ablative variants across all the three metrics we have, reducing the GT split ratio by 50% on average, as shown in Table 1b.

On PartNet-Chairs, LORL also helps both MONet and Slot Attention improve with a large margin, as shown in Table 2. The results are averaged over 4 runs. MONet in general performs well on this dataset, though it still sometimes merges different parts with the same color into a single mask. An example can be found in Fig. 5, column 3, where the blue arm and the blue bottom in the input image are put into the same mask by MONet. Such an issue is alleviated in LORL + MONet. Fig. 5 also includes examples to show how LORL helps Slot Attention.

As shown in Table 2, the improvement on Slot Attention is larger and more consistent, compared with the improvement on MONet. We hypothesize that this is because the two models adopt different approaches for aligning object features and masks. While MONet uses separate modules for segmentation and object representation learning, Slot Attention obtains masks by directly decoding

|  | $k = 1$ | $k = 3$ | $k = 5$ |
|---|---|---|---|
| Slot Attention | 54.07±2.6 | 43.70±1.7 | 37.02±1.7 |
| LORL + SA | **94.03±0.6** | **91.03±1.3** | **87.71±1.7** |

|  | QA Accuracy |
|---|---|
| LORL + SA (No FT) | 62.79±1.6 |
| LORL + SA | **92.72±1.0** |

Table 3: The percentage (%) of retrieved objects that belong to the same category as the query object. With LORL, objects within the same category are more likely to be close to each other in the feature space. The results are averaged over 3 runs.

Table 4: Our three-stage training paradigm improves visual concept learning. Without fine-tuning (i.e., the third training stage), the question answering accuracy drops by 30%. The results are averaged over 3 runs.

object representations. Having a shared representation might have allowed Slot Attention to gain more from language supervision.

## 5.2 INSTANCE RETRIEVAL

We now analyze the learned object representations on Shop-VRB-Simple. We first use them for instance retrieval: for each model, we randomly select a segmented object and use its learned representation to search for its $k$ nearest neighbors in the feature space. Then, for each selected object, we compute how many of the $k$ nearest neighbors belong to the same category. During searching, we only consider object representations whose corresponding mask, after decoding, has at least an Intersection over Union (IoU) of at least $0.75$ with a ground truth object mask. We sample 1,000 object features from each model for evaluation.

Table 3 includes results with $k = 1, 3, 5$, suggesting that the object representations learned by LORL + Slot Attention are better for retrieval, compared with features learned by Slot Attention alone without language. This is because Slot Attention often confuses categories that are visually similar but conceptually different, such as *baking tray* and *chopping board*.

## 5.3 VISUAL REASONING

As another analysis, we also evaluate how the learned representation of LORL +Slot Attention performs on visual question answering on the Shop-VRB-Simple dataset. Here we compare with an ablated version of LORL, where we only train the model for the first two stages, as stated at the end of Section 4. We do not train the model for the third stage—jointly optimizing or fine-tuning all three trainable modules. We name this ablation LORL + SA (No FT). Through this analysis, we hope to understand the importance of joint training of the vision modules (Modules 1 and 2) and the reasoning module (Module 4).

Table 4 shows that joint training is crucial for visual reasoning. This resonates with the previous result, where visually similar objects are clustered together in the latent space, impeding the usefulness of the information encoded.

## 5.4 REFERRING EXPRESSION COMPREHENSION

Finally we evaluate the representations learned by LORL on referring expression comprehension, where given an expression referring to a set of objects in the scene, like "The white plates", the model is expected to return all of the corresponding object masks. After learning all needed concepts from question-answer pairs, LORL can naturally handle referring expression without any further training, if we assume a pre-trained semantic parser.

We choose the IEP-Ref model (Liu et al., 2019) as our baseline. It uses a module approach and receives direct segmentation supervision. On Shop-VRB, we adapt the code provided by Liu et al. (2019) to generate 17K training examples, only for IEP-Ref, and 1.7K testing referring expressions for evaluating both LORL and IEP-Ref. Measured in Recall@0.5 (the ratio of the recalled objects based on an IoU threshold of 0.5), IEP-Ref performs better than LORL, but the margin is small (90.1% *vs*. 84.4%). Note that while IEP-Ref has been trained on 17K training examples with ground truth object segmentations, LORL does not require any training data on referring expression comprehension. The relatively comparable results are strong evidence that the representations learned by LORL also transfer to a new task.

## 6 CONCLUSION

We have proposed Language-mediated, Object-centric Representation Learning (LORL), a paradigm for learning object-centric representations from vision and language. Experiments on Shop-VRB-Simple and PartNet-Chairs show that language significantly contributes to learning better representations. This behavior is consistent across two unsupervised image segmentation models.

Through systematic studies, we have also shown how LORL helps models to learn object representations that encode conceptual information, and are useful for downstream tasks such as retrieval, visual reasoning, and referring expression comprehension.

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

## A    DOMAIN-SPECIFIC LANGUAGE (DSL)

LORL extends the domain-specific language of the CLEVR dataset (Johnson et al., 2017a) to accommodate descriptive sentences. Specifically, we add an extra primitive operation: `Equal(X, y)`. It takes two inputs. In our case, the first argument `X` is the output of a `Query`, `Exist`, or `Count` operation. All three operations output a distribution over possible answers. The second argument `y` is either a word or number, such as `TRUE`, *white*, or 4. The `Equal` operation computes the probability of `X=y`. In LORL, models are trained to maximize the output probability.

## B    HYPERPARAMETERS

For optimization hyperparameters, we largely adopt original settings in Burgess et al. (2019) and Locatello et al. (2020). Table 5 summarizes the hyperparameters for the loss weights ($\alpha$ and $\beta$), the number of training epochs of different stages ($N_1$, $N_2$, $N_3$), and the batch size. We early-stop the training when QA performance converges. We were skipping the second training phase on PartNet-Chairs because the first training phase (vision-only) yields very poor segmentation performance on this dataset. Establishing a meaningful grounding of concepts could be hard in this case. If we keep the second training phase for LORL + Slot Attention on PartNet-Chairs, the model converges slower in the third training phase (15 more epochs in our experiments), but the final performance remains the same.

|  | Shop-VRB-Simple Slot Attention | PartNet-Chairs MONet | PartNet-Chairs Slot Attention |
|---|---|---|---|
| $\alpha$ | 1 | 0.01 | 1 |
| $\beta$ | 0.1 | 1 | 0.1 |
| Batch Size | 64 | 64 | 64 |
| $N_1$ | 800 | 800 | 400 |
| $N_2$ | 20 | 0 | 0 |
| $N_3$ | 80 | 200 | 200 |

Table 5: Hyperparameters of LORL

**Learning rate scheduling**   For Slot Attention models, during the first training stage (perception-only), we use the learning rate schedule described in the original paper on both datasets. Specifically, the initial learning rate is $4 \times 10^{-4}$, and is fixed 10K iterations. After that, we decay the learning rate by 0.5 for every 100K iterations. On PartNet-Chairs, after the first stage, Slot Attention models continue to use the same learning rate scheduling. For Shop-VRB-Simple, we switch to a fixed learning rate of 0.001 during $N_2$ phase, which takes 20 epochs. After 20 epochs, we decrease the learning rate to $2 \times 10^{-4}$. We further decrease the learning rate to $2 \times 10^{-5}$ after another 65 epochs. We use the Adam optimizer (Kingma & Ba, 2015) for Slot Attention models.

For MONet models, we use RMSProp with a learning rate of 0.01 during the first stage, and use 0.001 for the second and the third stage.

Meanwhile, we also follow NS-CL (Mao et al., 2019) to use curriculum learning. Specifically, in the second training stage, we limit the number of objects in the scene to be 3. In the third training stage, we gradually increase the number of objects in the scene and the complexity of the questions.

## C    PARTNET-CHAIRS DATASET

Table 6 illustrates all templates that we use to generate descriptive sentences on the PartNet-Chairs dataset. We also show the corresponding latent programs for each template.

In PartNet-Chairs, all programs for descriptive sentences end with an `Equal` operator, which evaluates to true iff. both of its arguments are equal to each other. We train our model with paired images and sentences, by maximizing the probability that the first argument equals to the second argument.

Table 6: All templates that we use to generate descriptive sentences on the PartNet-Chairs dataset. The question mark indicates that the attribute preceding it is optional. "X/Y" means choosing either X or Y.

| Template | Example Sentence | Example Program |
|---|---|---|
| There is/are <number> <color>? (<part_name>/part). | There are 4 legs. | Scene, Filter[Leg], Count, Equal[4] |
| The number of <color>? (<part_name>/part) there is <number>. | The number of legs there is 4. | Scene, Filter[Leg], Count, Equal[4] |
| There is a <color>? ( <part_name>/part). | There is a brown seat. | Scene, Filter[Brown, Seat], Exist, Equal[True] |
| There is no <color>? ( <part_name>/part). | There is no brown seat. | Scene, Filter[Brown, Seat], Exist, Equal[False] |
| A <color>? ( <part_name>/part) is visible. | A brown seat is visible. | Scene, Filter[Brown, Seat], Exist, Equal[True] |
| No <color>? ( <part_name>/part) is visible. | No brown seat is visible | Scene, Filter[Brown, Seat], Exist, Equal[False] |
| The name of the <color> part is <part_name>. | The name of the yellow part is arm. | Scene, Filter[Yellow], Query[Part_Name, Equal[Arm] |
| The <color> part is called <part_name>. | The yellow part is called arm | Scene, Filter[Yellow], Query[Part_Name, Equal[Arm] |
| There is a <color> part; its name is <part_name>. | There is a yellow part; its name is arm. | Scene, Filter[Yellow], Query[Part_Name, Equal[Arm] |
| There is a <color> thing called <part_name>. | There is a yellow part called arm. | Scene, Filter[Yellow], Query[Part_Name, Equal[Arm] |
| The color of the <part_name> is <color>. | The color of the back is purple. | Scene, Filter[Back], Query[Color], Equal[Purple] |
| The <part_name> has <color> color. | The back has purple color. | Scene, Filter[Back], Query[Color], Equal[Purple] |

Table 6 (Continued)

| Template | Example Sentence | Latent Program |
|---|---|---|
| There is a <part_name>; its color is <color>. | There is a back; its color is purple. | Scene, Filter[Back], Query[Color], Equal[Purple] |
| The <part_name> is <color>. | The back is purple. | Scene, Filter[Back], Query[Color], Equal[Purple] |
| There is/are <number> <color>? (<part_name>/part) <relation> the <color>? ( <part_name>/part). | There is a green seat and a yellow leg is behind it. | Scene, Filter[Green, Seat], Unique, Relate[Behind], Filter[Yellow, Leg], Exist, Equal[True] |
| There is/are <number> <color>? (<part_name>/part) <relation> the <color>? ( <part_name>/part). | There is a green seat and no yellow leg is behind it. | Scene, Filter[Green, Seat], Unique, Relate[Behind], Filter[Yellow, Leg], Exist, Equal[False] |
| <number> <color>? (<part_name>/part) is/are <relation> the <color>? ( <part_name>/part). | 0 leg is behind the brown leg. | Scene, Filter[Brown, Leg], Unqiue, Relate[behind], Filter[Leg], Count, Equal[0] |
| There is a <color>? ( <part_name>/part); <number> <color> <part_name> is/are <relation> it. | There is a brown leg; 0 leg is behind it. | Scene, Filter[Brown, Leg], Unqiue, Relate[behind], Filter[Leg], Count, Equal[0] |
| The number of <color>? (<part_name> /part) that is/are <relation> <color>? (<part_name>/part)is <number>. | The number of leg that is behind the brown leg is 0. | Scene, Filter[Brown, Leg], Unqiue, Relate[behind], Filter[Leg], Count, Equal[0] |
| There is a <color>? (<part_name> /part) <relation> the <color>? (<part_name> /part). | There is a blue back behind the seat. | Scene, Filter[Seat], Unqiue, Filter[Blue, Back], Relate[behind], Exist, Equal[True] |
| There is no <color>?<part_name> <relation> the <color>? ( <part_name>/part). | There is no blue back to the right of the seat. | Scene, Filter[Seat], Unqiue, Relate[right], Filter[Blue, Back], Exist, Equal[False] |

Table 6 (Continued)

| Template | Example Sentence | Latent Program |
|---|---|---|
| There is a `<color>`? ( `<part_name>`/part) and a `<color>`? ( `<part_name>`/part) is `<relation>` it. | There is a seat and a blue back is to the right of it. | Scene, Filter[Seat], Unqiue, Relate[Right], Filter[Blue, Back], Exist, Equal[True] |
| There is a `<color>`? ( `<part_name>`/part); there is no `<color>`? `<part_name>` `<relation>` it. | There is a green seat; there is no blue back to the right of it. | Scene, Filter[Seat], Unqiue, Relate[Right], Filter[Blue, Back], Exist, Equal[False] |
| The color of the `<part_name>` (that is)? `<relation>` the `<color>`? ( `<part_name>`/part) is `<color>`. | The color of the leg left of the red arm is blue. | Scene, Filter[Red, Arm], Unique, Relate[Left], Filter[Leg], Query[Color], Relate[behind], Equal[Blue] |
| There is a `<part_name>` `<relation>` the `<color>`? ( `<part_name>`/part); its color is `<color>` | There is a leg to the left of the red arm; its color is blue. | Scene, Filter[Red, Arm], Unique, Relate[Left], Filter[Leg], Query[Color], Equal[Blue] |
| The name of the `<color>`? ( `<part_name>`/part) (that is)? `<relation>` the `<color>`? ( `<part_name>`/part) is `<part_name>`. | The name of the cyan part that is in front of the yellow seat is leg. | Scene, Filter[Yellow, Seat], Unique, Relate[Front], Filter[Cyan], Query[Part_Name, Equal[Leg] |
| The `<color>`? ( `<part_name>`/part) (that is)? `<relation>` the `<color>`? ( `<part_name>`/part) is called `<part_name>`. | The cyan part that is in front of the yellow seat is called leg. | Scene, Filter[Yellow, Seat], Unique, Relate[Front], Filter[Cyan], Query[Part_Name, Equal[Leg] |

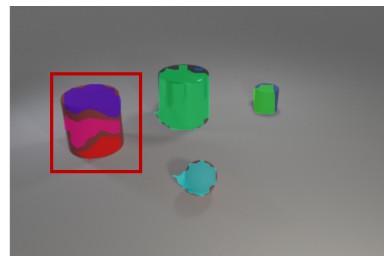 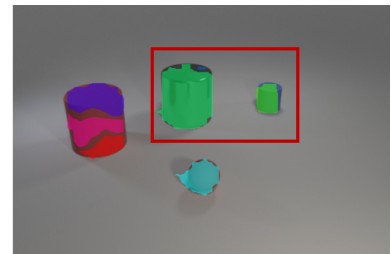

(a) There are 4 objects in the scene, and one of them is split into 3 masks. So GT Split = 1/4 = 0.25.

(b) There are 5 masks in the scene, and one of them covers two objects. So Pred Split = 1/5 = 0.2

Figure 6: A simple example on how GT/Pred Ratios are computed

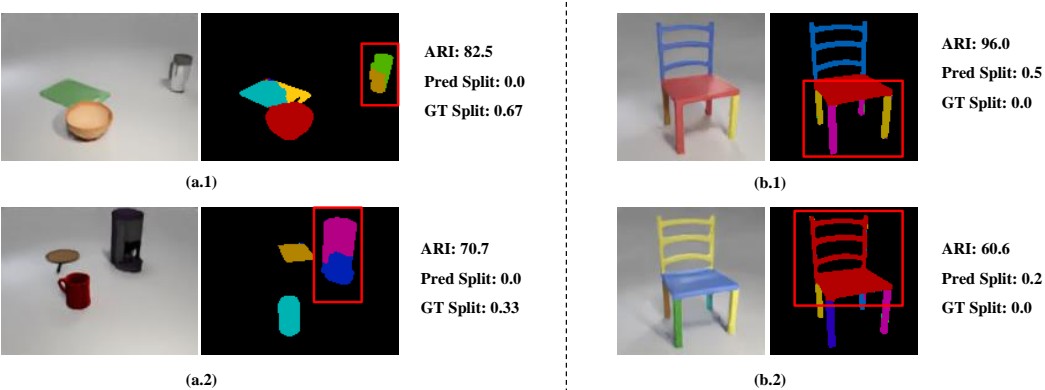

Figure 7: Comparison of ARI and GT/Pred Split Ratio on individual images. Pixels with the same color represent a mask produced by the models. We can see ARI is very sensitive to the size of objects, while Split Ratios capture object level failures where an object is split or multiple objects are merged.

## D    IMPLEMENTATION DETAILS

In this section, we provide additional implementation details of our experimental setups and metrics.

**GT and Pred split ratios.**    In this paper, we have introduced two new metrics for evaluating the performance of unsupervised object segmentation: namely, the GT split ratio and and the Pred split ratio.

A simple example of how we can compute GT/Pred split ratios is shown in Fig. 6. At a high level, the GT split ratio computes the percentage of objects that are split into multiple parts in model segmentation. Meanwhile, Pred split ratio computes the percentage of objects that are merged into a single object in model segmentation. We introduce these two new metrics because the ARI score is evaluated at the pixel level and does not account for the variance of object sizes. By contrast, GT split and Pred split metrics are computed at the object level. This difference is illustrated in Fig. 7.

For concrete examples, in the Fig. 6 (a.1), two objects, the chopping board and the thermos, are wrongly segmented. In Fig. 6 (a.2), only one object mis-segmented. However, the ARI score of the first image is much higher because the coffee maker has a large size. GT split ratio is evaluated on the object level and thus favor the second one. Similarly, in the Fig. 6 (b.1) , the four legs are merged into two masks, while in Fig. 6 (b.2), the seat and the back of the chair are merged into a single object. However, the first segmentation result has a significantly higher ARI score because the chair legs only contribute to a small area in the image. In this paper, we propose to jointly use ARI scores and the proposed GT/Pred split ratio to evaluate segmentation masks.

|  | Thres = 0.1 | | Thres = 0.3 | |
|---|---|---|---|---|
|  | GT Split | Pred Split | GT Split | Pred Split |
| SA (Locatello et al., 2020) | 23.71±1.5 | 23.45±4.1 | 8.91±0.7 | 7.47±0.9 |
| LORL + SA | **17.74±1.4** | **18.15±1.2** | **5.86±1.3** | **5.67±0.7** |

Table 7: GT/Pred split ratios on Shop-VRB-Simple using different IoU thresholds. The results are averaged over 3 runs.

|  | ARI ↑ | GT Split ↓ | Pred Split ↓ |
|---|---|---|---|
| SA (Locatello et al., 2020) | **83.51±2.3** | **15.68±1.9** | 13.19±1.5 |
| SA + Obj score | 83.4±1.8 | 16.57±1.5 | **12.71±0.8** |

Table 8: Ablation study of the objectness score module on Shop-VRB-Simple. The results are averaged over 3 runs.

Throughout the paper, we have been using IoU=0.2 as the threshold while computing the GT/Pred split ratios. Table 7 summarizes the results with different IoU thresholds. LORL consistently improves the baseline.

**Referring expression comprehension.** In this experiment, the data is generated using the code adapted from Liu et al. (2019). It contains two types of expressions, the first one directly refers to an object by its properties: for example, "the white plate". The second type of sentences refers to the object by relating it with another object: for example, "the object that is in front of the mug." The output of the model is the masks of all referred objects. The dataset is composed of the same set of concepts and same DSL as in the Shop-VRB-Simple.

We use the IEP-Ref model proposed in Liu et al. (2019) as a baseline. It is adapted from its prior work IEP (Johnson et al., 2017b). IEP-Ref first translates the referring expression into a sequence of primitive operations, which are implemented as different modular networks. The model takes the image feature extracted by a CNN as input and executes the program by chaining these component networks. It outputs a segmentation mask on objects being referred to. During training, groundtruth segmentation masks are needed.

For all methods, including ours and the baseline, we assume a pretrained semantic parser. Since the neuro-symbolic program executor outputs a distribution over all objects indicating whether they are selected, we directly multiply its output with the object segmentation masks to get the final output.

## E  ADDITIONAL RESULTS

The following section presents a collection of ablation studies on different modules of LORL, as well as a few extensions.

**Objectness score.** To validate the effectiveness of the proposed objectness score module, we hereby compare two models: the original Slot Attention model and the Slot Attention model augmented with the proposed objectness score module. Both models are trained using images only (there is no language in the loop), on the Shop-VRB-Simple dataset. The Table 8 summarizes the result. The objectness module alone does not contribute to the segmentation performance.

**Question type.** In this section, we investigate how different types of questions affect LORL. We use the Shop-VRB-Simple dataset for evaluation. There are three types of questions in the dataset, counting (e.g., how many plates are there?), existence (e.g., is there a toaster?), and query (e.g., what is the color of the mug?). We train LORL +SA with only one single type of questions (the number of total questions is the same).

|  | ARI ↑ | GT Split ↓ | Pred Split ↓ |
|---|---|---|---|
| SA (image-only) | 83.51±2.3 | 15.68±1.9 | 13.19±1.5 |
| Count only | 85.52±1.7 | 13.86±2.0 | 12.56±1.4 |
| Exist only | 86.29±2.2 | 15.34±2.9 | 10.33±1.1 |
| Query only | 88.79±1.1 | 10.64±1.7 | 9.03±0.5 |
| All types | **89.23±1.6** | 9.95±1.6 | **10.18±1.3** |

Table 9: Ablation study of using different types of questions to train LORL + SA on Shop-VRB-Simple. The results are averaged over 3 runs.

|  | ARI ↑ | GT Split ↓ | Pred Split ↓ |
|---|---|---|---|
| 25% (22.5K) | 81.01 | 14.31 | 15.34 |
| 50% (45K) | 84.39 | 14.79 | 9.37 |
| 75% (67.5K) | 86.53 | 11.67 | 11.52 |
| 100% (90K) | 89.23 | 9.95 | 10.18 |

Table 10: Ablation study of using different number of questions to train LORL + SA on Shop-VRB-Simple.

Results are summarized in Table 9. In general, training on all three types of questions improves the segmentation accuracy. The largest gain comes from the query question. Interestingly, the best result is achieved when trained on the original dataset, where the ratio of counting, existence, and query questions is 1:1:7. Note that all these models are trained with the same number of questions and thus they are directly comparable with each other.

**Data efficiency.** In addition, we provide another analysis by comparing models trained with different number of question-answer pairs. The results are shown in Table 10. Adding more language data consistently improves the result. All results are based on the LORL +SA model trained on the Shop-VRB-Simple dataset.

**Visual reasoning baselines.** To establish a baseline for visual reasoning tasks, we present the results of two visual reasoning approaches IEP (Johnson et al., 2017b) and NS-CL (Mao et al., 2019) for reference on the Shop-VRB-Simple dataset, as shown in Table 11. All models are trained with the same set of question-answer pairs. Note that NSCL has the access to a pretrained object detection module, while LORL +SA and IEP do not. LORL +SA outperforms IEP, which is trained with exactly the same amount of supervision as ours. It also achieves a comparable result as NS-CL.

**Integration with SPACE.** SPACE (Lin et al., 2020) is another popular method for unsupervised object-centric representation learning. SPACE uses parallel spatial attention to decompose the input scene into a collection of objects, and it is also compatible with the proposed learning paradigm LORL. We include additional results of LORL +SPACE on the CLEVR dataset. Shown in the Table 12, LORL +SPACE shows a significant advantage over the vanilla SPACE model. Additionally, we find that SPACE shows poor segmentation results on Shop-VRB-Simple and ParNet-Chairs, no matter whether it is integrated with LORL. For example, it frequently segments complex objects into too many fragments on Shop-VRB-Simple. We conjecture that this is because SPACE was designed for segmenting objects of similar sizes.

**Baseline using language supervision.** We also conducted an additional baseline model that uses language supervision in a different way. Specifically, based on the Slot Attention model, we use a GRU to directly encode question and answer, and concatenate it with the image feature to obtain the

|  | QA Accuracy |
|---|---|
| LORL + SA (No FT) | 62.79±1.6 |
| IEP (Johnson et al., 2017b) | 78.3±0.1 |
| NSCL (Mao et al., 2019) | **97.9±0.0** |
| LORL + SA | 92.72±1.0 |

Table 11: Question answering accuracy of baseline models and our model on the Shop-VRB-Simple dataset. The results are averaged over 3 runs.

|  | ARI ↑ | GT Split ↓ | Pred Split ↓ |
|---|---|---|---|
| SPACE (Lin et al., 2020) | 72.34 | 29.2 | 12.38 |
| LORL + SPACE | **97.82** | **2.04** | **2.17** |

Table 12: Segmentation performance of SPACE and LORL +SPACE on CLEVR. The integration of LORL improves the result.

|  | Precision | | | Recall | | |
|---|---|---|---|---|---|---|
|  | @0.5 | @0.75 | @1 | @0.5 | @0.75 | @1 |
| LORL + SA (NO) | 59.52±0.7 | 54.13±2.2 | 36.36±3.6 | **94.96±1.2** | **86.4±3.6** | **58.07±5.7** |
| LORL + SA | **89.47±0.7** | **79.37±1.2** | **52.48±1.6** | 94.72±0.1 | 84.02±0.6 | 55.55±1.2 |

Table 13: Concept quantification evaluation. The number after @ indicates the IoU threshold. The results suggest that objectness score improves the precision of concept quantification.The results are averaged over 3 runs.

object representation. On Shop-VRB-Simple, this model does not show improvement over the Slot Attention baseline: ARI=76.4%, GT Split=29.2%, Pred Split=13.6%. This suggests the effectiveness of LORL.

**Concept quantification.** Although LORL without the objectness score can achieve a comparable result in terms of QA accuracy, objectness score is crucial if we want to evaluate how models discover objects in images. Here, we show that, on the Shop-VRB-Simple dataset, LORL +SA shows significant improvement in recovering a holistic scene representation.

Specifically, we extract a scene graph for each scene, where each node corresponds to a detected object. We represent each node $i$ as a set of concepts $C_i$ associated with the object (e.g., {*large, brown, wooden, chopping board*}). We associate a concept with a detected object if its cosine similarity with the object representation is greater than 0. We heuristically remove nodes that are not associated with any concepts (by treating them as "background" objects) or have objectness scores that are smaller than 0.5. This results in a scene graph, where each node corresponds to a detected object. In the following, we compare it against the groundtruth scene graph.

For each pair of groundtruth node $i$ and detected node $j$, we compute the concept IoU score based on their associated concepts $C_i$ and $C_j$ as:

$$IoU_{ij} = \frac{|C_i \cap C_j|}{|C_i \cup C_j|}$$

Next, we perform a maximum weight matching between the detected scene graph and the groundtruth scene graph with the Hungarian algorithm. We use the IoU score as the weight for every edge and remove edges whose IoU score is smaller than a given threshold. Finally, based on the matching, we

can compute the precision and recall of the detected scene graph. We show the average precision and recall over the entire test set images in Table 13. The results suggest that objectness score significantly improves the precision of the extracted concepts.

