# OpenReview forum: "Language-Mediated, Object-Centric Representation Learning"
_ICLR.cc/2021/Conference — Reject_

### Official Review · AnonReviewer3 · 2020-10-20
**Interesting direction, but limited novelty of current approach and unsurprising results**

**Rating:** 4
**Confidence:** 4

**Review:**

### Summary

This paper proposes to combine the neuro-symbolic concept learner for visual reasoning from language (NS-CL; Mao et al., 2019) with recent unsupervised approaches to learning object-centric representations such as MONet (Burgess et al., 2019) and Slot-Attention (Locatello et al., 2020). While NS-CL normally relies on pre-trained object-detectors (in a supervised fashion) to extract visual representations, the proposed combination (dubbed LORL) use MONet or Slot Attention for this. By additionally back-propagating error signals from language-driven visual reasoning tasks obtained via NS-CL into MONet/Slot-Attention, it is shown how LORL is better able at learning object-centric representations and perform instance segmentation.

### Pro’s / Con’s / Justification

Overall the paper is well written and easy to follow. Moreover, using other modalities such as language to improve visual perception is an interesting research direction and timely due to recent advances in object-centric representation learning

However, the significance of this contribution is rather limited due to two main reasons:

LORL is a very straightforward combination of MONet/Slot-Attention with NS-CL and therefore not very novel --- the encoder part of NS-CL is simply replaced by the MONet/Slot-Attention, while everything else remains pretty much the same. Note that I don’t consider the “objectness scores” to modulate the NS-CL reasoning process very novel, since it is essentially a straight-forward heuristic to cope with a limitation of MONet/Slot-Attention in that sometimes the object-centric representations may containing background information or are “empty”. Were object-centric representations correctly inferred (as in the original NS-CL), this heuristic would therefore also not be needed

The results are not very surprising: it is found that better object-centric representations can be learned by fine-tuning on a visual reasoning task that uses language. However, since this task is learned in a supervised fashion and the dataset contains questions of the form “what is the name of the white object?” (parsed by a pre-trained semantic parser using a custom DSL), this provides a substantial degree of supervision to the representation learning part. In general, it is then not very surprising that using supervised data for fine-tuning improves representation learning, which limits the significance of this contribution further.

Due to these reasons, I can only recommend rejection at this point in time.

### Potential Improvements

I don’t think that it is straightforward to improve the current submission, although I have some detailed comments that I would like to see addressed below.

More generally I would encourage the authors to see if visual reasoning _alone_ (i.e. running solely the fine-tuning stage, but without the perception loss) would cause object-centric representations to emerge. Firstly, in that case LORL would not have to rely so much on existing contributions from prior approaches like MONet/Slot-Attention (a perceptual loss for which we know it already yields reasonably good object-centric representations). More importantly, in this case it is less obvious that a supervised language-based approach would work, since there is no guaranteed separation into approximate object-representations due to using a perception loss. This would almost certainly make the contribution more significant.

It would also be interesting to analyze in what situations fine-tuning on VQA is helpful and whether there are certain types of questions that are particularly helpful in this case. I could image that questions focusing on individual object properties are more useful than those that focus on more global information (like the total number of ceramic objects). More generally it would be interesting to quantify the amount of supervision given to the representation learning and comparing this to the amount of improvement gained.

### Detailed comments

* Regarding the objectness score an ablation is missing. What is the effect of this when learning object-centric representations? I expect it to be marginal, but it would still be good to demonstrate this. It would also be good to add the reasoning accuracy when the objectness modulation is not used to Table 4.

* Similarly, is it correct that the baseline approach in Tables 1a and 1b includes the objectness score, and is thus used for the proceeding training stages to arrive at the results for LORL? If not then please add to these tables the performance of LORL after the perceptual training phase (and if yes, please add a baseline that does not include the objectness score as per my previous comment)

* The standard deviation in Tables 1b and 2 is generally quite high, which makes it difficult to compare in some cases (although I agree that LORL generally outperforms the purely perceptual approach as is also expected). Therefore I would ask that you add additional seeds at least for Table 2 to allow for a better comparison.

* The proposed metrics (GT Split and Pred Split) seem rather arbitrary and I don’t think provide much additional insight. The ARI score is consistently in favor of LORL, and although the magnitude of the difference when comparing is sometimes greater when using GT Split and Pred Split it is unclear to me how to interpret that magnitude. More generally, the choice of assigning a mask to an object if it claims at least 20% currently seems arbitrary, while this could be an important hyper-parameter for these losses. For example, what do the results look like when using a threshold of 10% or 30%?

* I don’t find the results in Table 3 very surprising, since isn’t this what LORL is exactly trained for given the kind of questions used for the fine-tuning stage? What else could explain the improved ARI and the results in Table 4 other than that the representations have become better for semantic retrieval? Perhaps I am missing some alternative interpretations.

* I would appreciate a comparison to NS-CT in Table 4, even though the latter is supervised. It would help understand how good the score obtained by LORL-SA is.

* Section 5.4 essentially feels like an after-thought and details are missing. It would be helpful if the baseline IEP-Ref approach could be explained in some more detail.

* Regarding the conclusion, I don’t see why LORL is a “principled framework”. What is the principle that is being applied here? And why is this desirable over other approaches?

* I noticed in Appendix B that on PartNet-Chairs the second training phase (where only the reasoning part is trained) is skipped. What is the reason for this? What happens if this is also done for the Shop-VRB-Simple dataset?

### Post Rebuttal

I have read the other reviews and the rebuttal. I appreciate the extensive revision and response of the authors. Indeed, several of the minor issues/clarifications that I had raised have now been addressed.

However, as noted in my initial review, my main concern with this work is the highly limited novelty and the significance of the findings:

* LORL is essentially a simple application of unsupervised object-centric representation learners (like MONet, Slot-attention) to the language-guided visual reasoning framework proposed in MAO et al (2019), where the pre-trained vision module is interchanged. As I have previously argued, the objectness score is a heuristic only needed to overcome a limitation of the considered vision modules, which is not very interesting. Indeed, this score is not needed for segmentation (which is the primary measure of success that the authors have adopted).
* The main finding, which is that providing some degree of supervision to purely unsupervised object-centric representation learners improves their performance, is not very surprising. Although one could argue that the visual reasoning task does not provide direct supervision on the object segmentations, the considered type of questions (and DSL) provide a sizable amount of supervision I believe (as is also evident from the observed fluctuations in Table 9).

One issue that I noticed in the revision is when comparing the results in Table 9 and Table 10. It can be seen how when training on the visual reasoning task using only 25% of the provided data (i.e. 22.5K as opposed to 90K) actually reduces segmentation performance, i.e. from 83.51 (image only) to 81.01. This is surprising, and perhaps somewhat concerning, since I would have expected any reasonable amount of supervision to be helpful and certainly not degrade performance. The authors do not provide an explanation for this behavior, while it appears to invalidate the main claim regarding the benefit of LORL in the general case.

For these reasons I remain in favor of a rejection.

---

> ### Author Response · Authors · 2020-11-25
> **Author Response (Part 1 of 3)**
>
> Thank you for your thoughtful reviews.
>
> **Q1: Importance of the objectness score.**
>
> A1: Please refer to our general response Q2 for additional ablation studies on the objectness score. First, most unsupervised object-centric representation learning methods studied in this paper (MONET [1] and Slot Attention [2]) output a predetermined number of objects for every image. Objectness score is crucial if we want to evaluate how models discover objects in images. Moreover, we show that the inclusion of objectness scores improves the precision of concept learning.
>
> **Q2: Object-centric representation in NS-CL.**
>
> A2: NS-CL [3] and the proposed LORL work on two different topics: NS-CL focuses on learning concepts by associating linguistic units with object representations. Thus, it assumes access to pre-trained object detectors. By contrast, our framework works on learning object-centric representation that jointly discovers objects from images and learns concepts from language. No annotations of object segmentation or pretrained models are needed.
>
> **Q3: Language as supervision for representation learning.**
>
> A3: Please refer to our general response for a clarification of our goal and contribution.
> Existing methods for unsupervised object representation learning are sensitive to hyperparameter choices and often fail when objects have complex geometry or have translucent parts. Actually, there are often ambiguities in object individuation: should we separate the seat and the back of a chair as individual objects or should they be considered as a single one? How concept learning from language can improve and mediate models’ individuation of objects in scenes has been largely underexplored, to the best of our knowledge.
>
> Moreover, we also conducted an additional baseline model that uses language supervision in a different way. Specifically, based on the Slot Attention model, we use a GRU to directly encode question and answer, and concatenate it with the image feature to obtain the object representation. On Shop-VRB-Simple, this model does not show improvement over the Slot Attention baseline: ARI=76.4%, GT Split=29.2%, Pred Split=13.6%.  This further suggests the effectiveness of our proposed integration of NS-CL and the objectness score module.
>
> **Q4: Experiment with only language supervision.**
>
> A4: Thanks for the great suggestion. We add a new experiment where we train the LORL + Slot Attention model without perceptual loss on Shop-VRB-Simple. Thus, this baseline is trained with visual reasoning supervision only. We found that QA accuracy gets stuck at a local optima of 52.5%. Intuitively, without good individuation of objects, it is difficult for the model to ground linguistic concepts onto the object representations. This shows the effectiveness of perceptual losses in unsupervised object-centric representation learning.
>
> **Q5: Question type ablation and data efficiency study.**
>
> A5: Thanks for your suggestion. There are three types of questions in our dataset, counting (e.g., how many plates are there?), existence (e.g., is there a toaster?), and query (e.g., what is the color of the mug?). Per request, we train LORL+SA on SHOP-VRB-Simple with only a single type of question (but the total number of questions is the same). Results are summarized in the following table.
>
> |            |   ARI  | GT split | Pred Split |
> |:----------:|:------:|:--------:|:----------:|
> | SA (image-only) | 83.46% |  16.13%  |   12.95%   |
> | only count | 85.52% |  13.86%  |   12.56%   |
> | only exist | 86.29% |  15.34%  |   10.33%   |
> | only query | 87.99% |  13.37%  |    8.75%   |
> |  all types | 89.23% |   9.95%  |   10.18%   |
>
> In general, training on all three types of questions improves the segmentation accuracy (compared with the image-only baseline). The largest gain comes from the query-typed questions. Interestingly, the best result is achieved when trained on the original dataset, where the ratio of counting, existence, and query-typed questions is 1:1:7. Note that all these models are trained with the same number of questions and thus they are directly comparable with each other.
>
> In addition, we provide another analysis by comparing models trained with a different number of question-answer pairs. The results are shown below. Adding more language data consistently improves the result. All results are based on the LORL+SA trained on the Shop-VRB-Simple dataset.
>
> |             |   ARI  | GT Split | Pred Split |
> |:-----------:|:------:|:--------:|:----------:|
> |  25% (22.5K)| 81.01% |  14.31%  |   15.34%   |
> |  50% (45K)  | 84.39% |  14.79%  |    9.37%   |
> |  75% (67.5K)| 86.53% |  11.67%  |   11.52%   |
> | 100% (90K) | 89.23% |   9.95%  |   10.18%   |

---

> > ### Author Response · Authors · 2020-11-25
> > **Author Response (Part 2 of 3)**
> >
> > **Q6: Variance in PartNet-Chairs experiments.**
> >
> > A6: Per request, we have updated table 2 to include the average performance of models trained with 4 random seeds.
> >
> > |            |      ARI      |    GT Split   |   Pred Split  |
> > |:----------:|:-------------:|:-------------:|:-------------:|
> > |    MONet   | 91.41 +- 3.68 | 10.30 +- 5.13 | 14.09 +- 5.16 |
> > | LORL+MONet | 94.91 +- 2.13 |  4.95 +- 0.74 |  4.02 +- 2.52 |
> > |     SA     | 87.32 +- 3.59 | 12.54 +- 6.59 | 22.99 +- 5.04 |
> > |  LORL + SA | 95.81 +- 0.96 |  3.39 +- 1.05 |  2.92 +- 0.97 |
> >
> > In general, we see larger variance of models trained without language supervision. This is because initialization plays an important role in the convergence of these models. Since a single number of variance does not fully capture the distribution of model performance, we strongly recommend the reviewers to look at the performance histograms of different models on https://sites.google.com/view/lorl-iclr/ (Figure 5).
> >
> > **Q7: GT Split and Pred Split:**
> >
> > A7: We have included additional explanation of the proposed metrics in our revision.
> >
> > First, at a high level, the GT Split computes the percentage of objects that are split into multiple parts in model segmentation. Meanwhile, Pred Split computes the percentage of objects that are merged into a single object in model segmentation. We introduce these two new metrics because the ARI score is evaluated at the pixel level and does not account for the variance of object sizes. By contrast, GT Split and Pred Split metrics are computed at the object level.
> >
> > Second, in our experiment, we have observed two typical failure modes that are shared by multiple baselines on both datasets: splitting one object into several masks, and merging multiple objects into a single mask. Please refer to Figure 1 on https://sites.google.com/view/lorl-iclr/ for more qualitative examples. These two metrics are designed to quantify these two failure modes.
> >
> > We have provided concrete examples to demonstrate how different metrics are computed in Section “Comparison of ARI and GT/Pred Split Ratios” on https://sites.google.com/view/lorl-iclr/, as well as in the updated appendix.
> >
> > We thank the reviewer for suggesting additional ablations on the threshold. The following tables show the results if we use different thresholds. The results are presented on the Shop-VRB-Simple dataset. LORL consistently improves the baseline.
> >
> > Threshold = 0.1
> >
> > |         | GT Split | Pred Split |
> > |:-------:|:--------:|:----------:|
> > |    SA   |  23.71%  |   23.45%   |
> > | LORL+SA |  17.74%  |   18.15%   |
> >
> > Threshold = 0.3
> >
> > |         | GT Split | Pred Split |
> > |:-------:|:--------:|:----------:|
> > |    SA   |   8.91%  |    7.47%   |
> > | LORL+SA |   5.86%  |    5.67%   |
> >
> > **Q8: Interpretation of Table 3.**
> >
> > A8: We have updated our text to include more discussions of Table 3. The main message of Table 3 is that, models trained with perceptual losses alone are able to capture object semantics to a certain degree, but worse than models trained with language supervision.
> >
> > **Q9: More baselines on visual reasoning tasks.**
> >
> > A9: We kindly refer the reviewer to our general response Q3.
> >
> > **Q10: Second phase of training.**
> >
> > A10: Thanks for the suggestion. We were skipping the second training phase on PartNet-Chairs because the first training phase (vision-only) yields very poor segmentation performance on this dataset. Establishing a meaningful grounding of concepts could be hard in this case. Per request, we have conducted a new ablation study. If we keep the second training phase for LORL + SlotAttention on PartNet-Chairs, the model converges slower in the third training phase (15 more epochs in our experiments), but the final performance remains the same.
> >
> > **Q11: Details of IEP-Ref.**
> >
> > A11: We have added these details to our revision. We use the IEP-Ref model proposed in [3]. It is adapted from its prior work IEP [4]. IEP-Ref first translates the referential expression into a sequence of primitive operations, which are implemented as different modular networks. The model takes the image feature extracted by a CNN as input and executes the program by chaining these component networks. It outputs a segmentation mask on objects being referred to. During training, groundtruth segmentation masks are needed.

---

> > > ### Author Response · Authors · 2020-11-25
> > > **Author Response (Part 3 of 3)**
> > >
> > >
> > > ```
> > > [1] Christopher P Burgess, Loïc Matthey, Nicholas Watters, Rishabh Kabra, I. Higgins, M. Botvinick and Alexander Lerchner. MONet: Unsupervised Scene Decomposition and Representation. arXiv:1901.11390.
> > > [2] Locatello, Francesco, Dirk Weissenborn, Thomas Unterthiner, Aravindh Mahendran, Georg Heigold, Jakob Uszkoreit, A. Dosovitskiy and Thomas Kipf. Object-Centric Learning with Slot Attention. In NeurIPS, 2020.
> > > [3] Runtao Liu, Chenxi Liu, Y. Bai and A. Yuille. CLEVR-Ref+: Diagnosing Visual Reasoning With Referring Expressions. In CVPR, 2019.
> > > [4] Justin Johnson, Bharath Hariharan, Laurens van der Maaten, Li Fei-Fei, C Lawrence Zitnick, and Ross Girshick. CLEVR: A Diagnostic Dataset for Compositional Language and Elementary Visual Reasoning. In CVPR, 2017.
> > > ```

---

### Official Review · AnonReviewer4 · 2020-10-28
**Intuitive approach mostly based on existing components, somewhat limited evaluation to support all the claims.**

**Rating:** 5
**Confidence:** 4

**Review:**

=Summary
The paper proposes a framework for object-centric representation learning with additional language supervision such as e.g. questions and answers, denoted as Language-mediated, Object-centric Representation Learning (LORL). The authors combine two ideas from prior work, the unsupervised object-centric representation learning and the neural-symbolic concept learning, in one architecture. The model obtains object representations by learning to reconstruct the input image (as in MONet and Slot Attention). The learned representations are used as input to the neural-symbolic program executor, which learns to answer questions about objects. The entire model is trained in three stages: first the reconstruction objective, then the QA objective, and, finally, jointly. Experiments on two datasets demonstrate that the obtained object segmentations have better quality that those of the original unsupervised models. The learned representations are also shown to be effective in several other down-stream tasks.

=Strengths

The intuitions are clearly conveyed and the method is well motivated: language introduces strong inductive biases, providing concept names which can in turn be grounded in images, helping the model understand what these concepts should look like (especially for heterogenous objects such as “coffee maker”).

The authors have collected language descriptions for the PartNet-Chairs dataset.

Adding LORL on top of Slot Attention leads to notable improvement in object segmentation performance on two datasets.

The learned representations (using Slot Attention) allow to retrieve similar objects that belong to the same object category, showing that they capture object semantics much better than the unsupervised models.

It is interesting to see that the learned representations can be effectively used for other tasks, e.g. referring expression comprehension w/o additional fine-tuning.

=Weaknesses/High-level comments

I am not sure how unexpected the main result really is (the improved segmentation quality), considering that LORL receives additional (language) supervision. Comparison of vanilla unsupervised models vs. LORL is not completely fair or informative, in my opinion. It would be interesting to compare LORL vs. some other forms of supervision (or even fully/partially supervised segmentation models) or show how VQA supervision compares to caption supervision (more on that below), or using attributes instead of language, etc.

The authors claim that “LORL can be integrated with various unsupervised segmentation algorithms” (P1). In practice almost all the experiments are carried out with just Slot Attention. The only reported result with MONet (Table 2) is somewhat weak (as also stated by the authors, P7). This seems to undermine the authors' claim.

There are no experiments on the popular CLEVR dataset, only on the two recent datasets introduced by the authors (ShopVRB-Simple is obtained by selecting easier scenarios from ShopVRB). Specifically, they could have used CLEVR for MONet, as MONet is not applicable to the ShopVRB-Simple data.

It is in unclear what the language task on the PartNet-Chairs actually is, it is only stated that the authors collected descriptive sentences. It is mentioned that the output can be True or False; does it mean that the task is to predict whether the sentence is true to the image? No”negative” examples are included in the paper, nor is there any discussion on that.

Why have the authors decided against collecting question-answering data for the PartNet-Chairs, similar to the ShopVRB-Simple? Is there an advantage of using a specific form of supervision (e.g. question-answering supervision)? Some analysis on which form of supervision is more effective would have been interesting.

The overall framework mostly relies on existing components, the main technical novelty is bringing them together and an additional objectness score used in both objectives. It is not entirely clear whether the reported Slot attention and MONet performance is obtained with the vanilla implementations or after stage 1 training of the proposed model (which also includes the objectness scores). Overall, would be interesting to see an ablation of the proposed objectness score.

Would be great to learn more about the referring expression comprehension experiment (what the test data looks like, how similar it is to the observed training data, how was the model adapted to the task, etc).

The authors show that the final training stage significantly improves the model’s QA abilities, saying that it demonstrates the importance of joint training. I am not entirely sure about the purpose of this experiment. Would be also interesting to see how the obtained performance compares to the vanilla neural-symbolic approach.

Here we have synthetic images with no background (relatively easy to segment) and synthetic (templated) language (easy to parse). I am somewhat skeptical about the generality of the proposed framework (which also concerns prior works on which this work builds), i.e. can we expect that this model will work with real cluttered images and real natural language? What do the authors believe is the main limitation of their proposed approach?

Nothing was stated about making the code available.

=Detailed comments (P# - page number)
- P1: perhaps “(for) learning disentangled, object-centric scene representations”?
- Throughout the paper the authors say “referential expression interpretation”, while this task is more commonly known as “referring expression comprehension”.
- The authors say “object” to refer to both objects (in ShopVRB-Simple) and object parts (in PartNet-Chairs). I wonder if there is a better way to explain this, which encapsulates both scenarios.
- Fig 1 is misleading as it misrepresents the task posed on the PartNet-Chairs dataset (it is not question-answering).
- P2 ShopVRB should be ShopVRB-Simple.
- P2 PartNet should be PartNet-Chairs.
- P3: should o_i be z_i?
- P3 white object Fig. 2 => in Fig. 2
- P4: should {z_i} be {z_k}?
- P4 which query the attribute => which queries the attribute
- P4 All concepts appear => All concepts that appear
- P4 “We use the same domain-specific language (DSL) for representing programs as CLEVR “ : in fact it is not the same but extended, as stated in the appendix.
- P5 an filter => a filter
- P5 “we find that it better highlights the quality of learned object-centric representations for various models.” - not clear what this means
- P7 “The quantitative results are summarized in Table 1.” - make it clear that this is for the ShopVRB-Simple

==================================================================

Post-rebuttal comments:

I thank the authors for an extensive response and the other reviewers for brining up many relevant questions!

= Main positive additions:

LORL was successfully combined with another unsupervised approach, SPACE, on the CLEVR dataset.
LORL+SA outperforms IET, which has the same amount of supervision. It does lose to NS-CL slightly, which has access to pre-trained object detectors.
I like the added analysis of the QA types, data efficiency etc.

= Some of the remaining issues:

The positive impact of the objectness score on performance was not demonstrated. To show its benefit the authors had to propose yet another evaluation scheme (precision and recall of the reconstructed scene graph).
The training objective for PartNet-Chairs should be discussed in the main paper, not in the appendix. Also, perhaps I am missing something, but would not one still need some negatives to train it?
Minor: Fig 1 in the revision is still wrong, i.e. the example for PartNet-Chairs dataset still illustrates the QA task.

Overall, I like that the approach is intuitive and well motivated but I also think the overall technical novelty is limited. The authors have considerably expanded their evaluation, but at the same time have introduced another confusion (as pointed out by R3 during post-rebuttal discussion): it appears that using 25% of supervision leads to lower segmentation performance, contradicting the main claim of the paper.
I therefore decrease my score to 5. I hope to see an improved version of the paper (with more exciting technical contributions) in a future venue!

---

> ### Author Response · Authors · 2020-11-25
> **Author Response**
>
> Thank you for your thoughtful reviews.
>
> **Q1: Language and other types of supervision.**
>
> A1: Please refer to our general response for a clarification of the goal and contribution. Existing methods for unsupervised object representation learning are sensitive to hyperparameter choices and often fail when objects have complex geometry or have translucent parts. Actually, there are often ambiguities in object individuation: should we separate the seat and the back of a chair as individual objects or should they be considered as a single one? How concept learning from language can improve and mediate models’ individuation of objects in scenes has been largely underexplored, to the best of our knowledge.
>
> We also agree with the reviewer that other forms of supervision, such as navigation and manipulation in interactive domains are meaningful and interesting supervisions as well. However, in this paper, we focus on how language can mediate object-centric learning. The inclusion of other types of supervisions and their comparisons remain future works.
>
> Per request, we have also included additional results to analyze how different types of questions and the number of question-answer pairs contribute to the LORL's performance. We kindly refer the reviewer to our Response to Reviewer 3 (Q5) for more details.
>
> **Q2: Integration with other unsupervised object discovery methods.**
>
> A2: We have included an additional experiment trying to integrate SPACE [3] with LORL. The results are shown on the CLEVR dataset. As shown in the table below, LORL+SPACE shows a significant advantage over the vanilla SPACE. Additionally, we find that SPACE shows poor segmentation results on Shop-VRB-Simple and ParNet-Chairs, no matter whether it is integrated with LORL. For example, it frequently segments complex objects into too many fragments on Shop-VRB-Simple. We conjecture that this is because SPACE was designed for segmenting objects of similar sizes.
>
> |            |   ARI  | GT Split | Pred Split |
> |:----------:|:------:|:--------:|:----------:|
> |    SPACE   | 72.34% |  29.20%  |   12.38%   |
> | LORL+SPACE | 97.82% |   2.04%  |    2.17%   |
>
> **Q3: MONet on CLEVR.**
>
> A3: Due to the time limit, we are unable to finish up running experiments on CLEVR. Based on the original paper [4], training MONET on CLEVR takes 2 weeks. However, it is worth noting that MONet has already achieved near perfect segmentation results on CLEVR, with an ARI of 96.2% [5]. This suggests that there is little room for future improvement by other integration.
>
> **Q4: Language task on PartNet-Chairs.**
>
> A4: Your interpretation is absolutely correct. In PartNet-Chairs, all programs for descriptive sentences end with an Equal operator, which evaluates to true iff. both of its arguments are equal to each other. We train LORL with paired images and sentences, by maximizing the probability that the first argument equals to the second argument.
>
> We use different text forms (questions vs. descriptive sentences) for different datasets to show the generality of the proposed method. Since question-answer pairs can usually be converted into descriptive sentences (e.g., what’s the color of the seat back => the color of the seat back is…), comparing question-based and descriptive sentence-based supervision is not the main focus of our study. For detailed study on different types of supervision, we kindly refer the reviewer to the “Question Type and Data Efficiency Study” in our response to R3 (Q5).
>
> **Q5: Objectness score ablation.**
>
> A5:  We kindly refer the reviewer to general response Q2 for an ablation study on the objectness score.
>
> **Q6: Referential expression comprehension details.**
>
> A6: Thanks for the suggestion. We have included the details of the referential expression task in the revision. In this experiment, the data is generated using the code adapted from [6]. It contains two types of expressions, the first one directly refers to an object by its properties: for example, “the white plate”. The second type of sentences refers to the object by relating it with another object: for example, “the object that is in front of the mug.” The output of the model is the masks of all referred objects. The dataset is composed of the same set of concepts and same DSL as in the Shop-VRB-Simple. For all methods, including ours and the baseline, we assume a pretrained semantic parser. Since the neuro-symbolic program executor outputs a distribution over all objects indicating whether they are selected, we directly multiply its output with the object segmentation masks to get the final output.

---

> > ### Author Response · Authors · 2020-11-25
> > **Author Response (Cont')**
> >
> > **Q7: Other visual reasoning baselines.**
> >
> > A7: We kindly refer the reviewer to general response A3 for more baselines on the visual reasoning task.
> >
> > **Q8: Real-world datasets.**
> >
> > A8: We agree with the reviewers that current experiment setups still can not match the real-world complexity. We believe that learning object-centric representation from vision, language, and other types of supervision such as interaction with objects is a meaningful direction. This paper has presented a unified approach for perceptual and language-mediated object representation learning, with successful application in domains composed of objects with complex geometries (Shop-VRB, PartNet). Concretely, we think the integration with more advanced models for complex object textures, backgrounds, and object dynamics [7,8] are meaningful future directions.
> >
> > ```
> > [1] Thomas Kipf, Elise van der Pol, and Max Welling. Contrastive Learning of Structured World Models. In ICLR, 2020.
> > [2] Bear, Daniel M., C. Fan, Damian Mrowca, Yunzhu Li, Seth Alter, Aran Nayebi, Jeremy I Schwartz, Li Fei-Fei, Jiajun Wu, J. Tenenbaum and D. Yamins.  Learning Physical Graph Representations from Visual Scenes. In NeurIPS, 2020.
> > [3] Zhixuan Lin, Yi-Fu Wu, S. Peri, Weihao Sun, G. Singh, Fei Deng, Jindong Jiang and Sungjin Ahn.  SPACE: Unsupervised Object-Oriented Scene Representation via Spatial Attention and Decomposition. In ICLR, 2020.
> > [4] Christopher P Burgess, Loïc Matthey, Nicholas Watters, Rishabh Kabra, I. Higgins, M. Botvinick and Alexander Lerchner. MONet: Unsupervised Scene Decomposition and Representation. arXiv:1901.11390.
> > [5] Klaus Greff, Raphaël Lopez Kaufman, Rishabh Kabra, Nick Watters, Christopher Burgess, Daniel Zoran, Loïc Matthey, M. Botvinick and Alexander Lerchner. Multi-Object Representation Learning with Iterative Variational Inference. In ICML, 2019.
> > [6] Runtao Liu, Chenxi Liu, Y. Bai and A. Yuille. CLEVR-Ref+: Diagnosing Visual Reasoning With Referring Expressions. In CVPR, 2019.
> > [7] Julian Ost, F. Mannan, N. Thürey, Julian Knodt and Felix Heide. Neural Scene Graphs for Dynamic Scenes. arXiv:2011.10379
> > [8] Rishi Veerapaneni, John D. Co-Reyes, Michael Chang, Michael Janner, Chelsea Finn, Jiajun Wu, Joshua B. Tenenbaum and Sergey Levine. Entity Abstraction in Visual Model-Based Reinforcement Learning. In CoRL, 2019.
> > ```

---

### Official Review · AnonReviewer1 · 2020-10-28
**Language mediated, Object-Centric Representation Learning**

**Rating:** 5
**Confidence:** 5

**Review:**

Focus of the work: The paper tries to tackle the problem of learning object centric representations using language. The authors note that most of the previous work which tries to use language  assume pre-trained object detectors to generate object proposals in the visual image.

Methodology: The proposed method consists of 4 modules.
1. Object Encoder, which tries to generate a modular representation of the scene.
2. Object Decoder, a module to reconstruct masks corresponding to individual objects given the high level factorial representation from the object encoder.
3. A pre-trained semantic parser: a pre-trained module to parse the input (for ex. a question) into semantically meaningful program which can be easily executed.
4. Neural-symbolic Program: This modules takes a factorial representation (i.e., the output of the encoder), as well as some other intermediate information from module 2 and module 2 and outputs an answer to the question.

The paper mentions that this is a "principled" framework for object centric learning. By keeping the module 3, and module 4 fixed, the authors evaluate different methods for learning object centric representations like MONET and SLOT attention. The authors evaluate how inclusion of the language can improve the performance of MONET and SLOT attention.

Experiments: The authors evaluate the proposed work on two visual reasoning datasets for image segmentation evaluation: Shop-VRB-Simple, and PartNet, as well as a subset of PartNet called PartNet-Chairs. The paper shows that inclusion of the language improves the performance of both MONET as well as SLOT attention, albeit the increase in performance corresponding to SLOT attention is more as compared to MONET.

Interesting points:

1. I like the problem which the proposed method hopes to achieve i.e., learning the representation of high level variables (i.e., objects), which are often associated with language.

2. It's interesting to note that the gain in performance for Slot attention is more as compared to MONET. This point was also emphasized in the work of RIMs (https://arxiv.org/abs/1909.10893), where they used a "top-down" representation to learn object centric representations.

3.  I like the ablation in table 4 i.e., fine-tuning the entire model improves the performance of the model, as compared to separately training the different components.

Major Points:

1.  I think the paper presents nice preliminary experiments for showing that incorporation of language can help learning object centric representations. Since the contribution of the paper is to actually evaluate different methods for object centric learning (MONET/SLOT Attention) and combining with the framework of neuro-symbolic learning. It would be interesting to evaluate the LORL + Slot Attention as well as LORL + MONET for more difficult tasks to see how well the learned representations transfer to new environments which share some common structure. It could be in the form of  language conditioned scene generation such that by using language one can generalize in a compositional way to new scenes or in the context of instruction following (instruction is in the form of language) where their are objects in the scene, and the goal is to put the objects in  a particular spatial configuration or just navigation (https://arxiv.org/abs/2003.05161).


Minor Comments:

"We have proposed Language-mediated, Object-centric Representation Learning (LORL), a principled
framework for learning object-centric representations from vision and language."

I'm not sure about the use of the word "principled framework" as the proposed method is not really a framework. As authors note in the paper, the goal of the paper is to see how incorporation of the language can be used to learn or improve representations of the high level variables (i.e., objects).

======

After Rebuttal: I have read the rebuttal, as well as reviews by other reviewers. I very much agree with the authors that the problem is very interesting, but as of now more work needs to be done in terms of "downstream applications".

---

> ### Author Response · Authors · 2020-11-25
> **Author Response**
>
> Thank you for your thoughtful review.
>
> **Q1: Evaluation of object-centric representation learning.**
>
> A1: Object-centric representation learning of images naturally integrates two components: 1) the discovery of objects in the image and 2) representation learning for individual objects. Thus, our experiment section starts with evaluating object discovery performance, measured by the pixel-level segmentation accuracy. This follows the standard evaluation pipeline in previous works [1, 2].
>
> Second, our model is able to learn object representations that are directly associated with concepts in latent embedding spaces. The modularized design of the neuro-symbolic concept learner [3] has enabled us to make interpretable quantification of object categories, colors, and other properties. As a result, the learned representation can be directly applied to downstream tasks such as visual question answering and referring expression comprehension. Given a specification of an object of interest in language, our model can directly produce the mask of the object. Such ability naturally transfers to other domains such as object manipulation or instruction following if we integrate our model with other planning algorithms, which we leave as future works.
>
> ```
> [1] Klaus Greff, Raphaël Lopez Kaufman, Rishabh Kabra, Nick Watters, Christopher Burgess, Daniel Zoran, Loïc Matthey, M. Botvinick, and Alexander Lerchner. Multi-Object Representation Learning with Iterative Variational Inference. In ICML, 2019.
> [2] Locatello, Francesco, Dirk Weissenborn, Thomas Unterthiner, Aravindh Mahendran, Georg Heigold, Jakob Uszkoreit, A. Dosovitskiy and Thomas Kipf. Object-Centric Learning with Slot Attention. In NeurIPS 2020.
> [3] Jiayuan Mao, Chuang Gan, Pushmeet Kohli, Joshua B. Tenenbaum, and Jiajun Wu. The Neuro-Symbolic Concept Learner: Interpreting Scenes, Words, and Sentences from Natural Supervision. In ICLR, 2019.
> ```

---

> > ### Comment · AnonReviewer1 · 2020-11-25
> > **Thanks! Similar concerns as other reviewers.**
> >
> > I thank the authors for taking time in writing a rebuttal, and running additional experiments.
> >
> > > ""However, our goal is not to improve existing methods, but to demonstrate how concept learning from language and unsupervised object discovery can bootstrap each other"
> >
> > Currently, I feel like the paper is not really written in a way which is aligned with the above goal.
> >
> > Just to reiterate I think, this is a great problem to study, as relation b/w different entities are normally sparse
> > (just like dependencies in language), which can be very useful for downstream tasks.
> >
> > Let's just take the example of the abstract.
> >
> >
> > > We present Language-mediated, Object-centric Representation Learning (LORL),
> > learning disentangled, object-centric scene representations from vision and language
> >
> > What is LORL ? LORL is not really a framework. The goal of the paper is to study how language and unsupervised learning of entities can bootstrap each other (as authors mentioned it themselves). In order to achieve this, the paper uses two methods for unsupervised object discovery. So, I think this is not an accurate description of what the paper is actually trying to do.
> >
> > > "But LORL further learns to associate the learned representations to concepts, i.e., words for object
> > categories, properties, and spatial relationships, from language input.
> >
> > Again, what really is LORL ? It's NOT a framework.
> >
> > > "These object centric concepts derived from language facilitate the learning of object-centric
> > representations"
> >
> > This line from the abstract accurately justifies what the paper is about. But the rest of the paper is not really written
> > in such a way.
> >
> > > "Experiments show that LORL consistently improves the performance of MONet and Slot Attention on two datasets via the
> > help of language."
> >
> > One can write it: Our experiments show that concept learning from language can improve the performance of XYZ... But again, the paper is over-claiming.
> >
> > > "We also show that concepts learned by LORL aid downstream tasks such as referential expression interpretation."
> >
> > Similarly here..
> >
> > I like the goal of the paper, but according to me exposition of the paper is not accurate as of now.
> >
> > > "Such ability naturally transfers to other domains such as object manipulation or instruction following if we integrate our model with other planning algorithms, which we leave as future works."
> >
> > It would be interesting to see, if the ability "naturally transfers". Its not as easy of a problem as authors seem to imply.
> >
> > I hope authors are doing well in such times.

---

> > > ### Author Response · Authors · 2020-11-25
> > > **Thank you for your thoughtful feedback!**
> > >
> > > Thank you for your prompt reply!  We appreciate your thoughtful suggestions and the chance to discuss with you.
> > >
> > > We realize the confusion might be due to different interpretations of the word “framework”. We agree with you that LORL is not about proposing another model/model, and in this sense not an “algorithmic framework”. Instead, as we both agree that the goal of LORL is to show “how language and unsupervised learning of entities can bootstrap each other”, we have treated it as a generic “learning framework” that can be integrated with multiple existing methods such as MONet, as we have shown in the paper.  To avoid confusion, we have revised the paper and referred to LORL as a “learning paradigm” instead.
> > >
> > > In the abstract, we used LORL to refer to the generic learning paradigm, not a specific model. Thus, when we say “But LORL further learns to associate the learned representations to concepts”, what we mean is the paradigm allows models such as MONet to learn to do so. The sentence “LORL consistently improves the performance of MONet“ indicates that the use of the learning paradigm makes MONet learn better.
> > >
> > > We’d like to clarify that we were not attempting to overclaim our contributions, because we have always interpreted “LORL” not as a specific model, but a general learning paradigm (or “learning framework” in the original version). But we agree that our original presentation might be ambiguous and confusing, and have thus revised the sentences accordingly.
> > >
> > > In terms of downstream applications, we have demonstrated that the learned concepts can be directly used in vision and language tasks such as interpreting referring expressions (Section 5.4). This highlights the value that LORL brings to methods such as MONet.
> > >
> > > Conceptually, such abilities may naturally serve as a prerequisite for downstream tasks in other disciplines, such as robot manipulation In practice, solving these tasks requires the integration of additional components, such as motion planners and controllers, which itself is a challenging research problem and a standalone research project.  This is why we believe it’s beyond the scope of this paper and leave it as future work.
> > >
> > > We sincerely hope our reply and the revised paper have better addressed your concerns. Thanks again for your effort in such times.
> > >
> > > Authors

---

### Official Review · AnonReviewer2 · 2020-11-03
**Interesting paper, but critical information is missing.**

**Rating:** 4
**Confidence:** 4

**Review:**

This paper proposed an interesting idea that uses language to learn the concept and aid downstream tasks such as segmentation and referential expression interpretation. The authors combine the unsupervised segmentation method (MONet and Slot Attention) with neural symbolic concept learning (NS-CL). By joint training these two objectives, the authors show improvements in the object segmentation and several downstream tasks.

1) What is the language objective for the PartNet-Chairs dataset? The dataset contains templated captions instead of questions, but the paper didn't mention any of the associate target or loss.

2) A lot of the paper needs more clarification, for example, how to calculate the  "Whiteness" using the concept embedding? How many programs are there? I can not find that information even in the supplementary materials. It is hard to understand the exact algorithm of the proposed method.

3) The performance of pretrained semantic parser is not reported in the paper. Since there are only a few templates, will the model achieve 100 percent performance?

4) The semantic parser info is necessary since the model will know the exact additional information of the image given the question and captions (especially captions), assuming there is no error in the parsing procedure. This will harm the overall novelty of the paper since the proposed method can not generalized to more complicated questions and captions.

5) If the pretrained semantic parser can provide very high accuracies, for a fair comparison, it will be good the authors can show the performance of MONet and Slot Attention with additional supervision. For example, Instead of having a Neuro-symbolic program executor, simply use the concept embedding to filter the object-centric representations and then do image decoding. This will show the effectiveness of the Neuro-Symbolic Program Executor.

6) Missing related work, [1] also learns the visual concept through question answering. The authors should discuss the difference between this work.
[1] Yang et.al Visual Curiosity: Learning to Ask Questions to Learn Visual Recognition (CORL)

I really like the paper's idea, but a lot of critical information experiments are missing. I am happy to increase my rating if the authors can resolve my questions.

---

> ### Author Response · Authors · 2020-11-25
> **Author Response**
>
> Thank you for your thoughtful review.
>
> **Q1: Training objective for the PartNet-Chairs dataset.**
>
> A1: We refer the reviewer to Appendix A and Table 6 for templates and example programs used in the PartNet-Chairs dataset. Specifically, all programs for descriptive sentences end with an Equal operator, which evaluates to true iff. both of its arguments are equal to each other. We train our model with paired images and sentences, by maximizing the probability that the first argument equals the second argument.
>
> **Q2: Computation of the “Whiteness” score.**
>
> A2: We use the same concept quantification module as [1]. Specifically, to compute the  “Whiteness” of an object, we first extract the object representation of the target object, project it into the concept space, and compute the cosine similarity between the object representation and a concept embedding for “white”.
>
> **Q3: The number of programs.**
>
> A3: On Shop-VRB-Simple, we use 10K images and 90K questions for training, 960 images, and 8.6K questions for testing. We use the same DSL as the original paper and thus there are 19 primitive operations. On PartNet-Chairs, there are 5K images and 40K captions for training, 960 images for testing; 10 different basic functions are considered.
>
> **Q3: The performance of the semantic parser.**
>
> A3: Thank you for the suggestion and we have included an evaluation of the semantic parsing module in our revision. Since the module is trained on paired question-program pairs, it achieves nearly perfect accuracy (>99.9%) on test questions/captions.
>
> **Q4: Generalization of the semantic parsing module to more complex questions.**
>
> A4: We agree with the reviewer that our current way of training semantic parsing modules has its limitations. However, in this paper, our main focus is on joint object discovery (segmentation) and concept learning. Parsing more complicated sentences is an orthogonal direction with our contribution. For example, potential future works may consider using semantic parsers trained on general corpus (e.g., [4]) or joint training of semantic parsing and concept learning [2].
>
> **Q5: The effectiveness of the Neuro-Symbolic Program Executor.**
>
> A5: The neuro-symbolic program executor indeed executes the program tokens, such as filter[White] by comparing the embedding of “white” with individual object representations. Per request, we have included an additional baseline, which uses a GRU to directly encode the questions and the answers associated with each image, and concatenate it with the image feature while extracting the representation for each object. On Shop-VRB-Simple, the modified Slot Attention model does not show improvement: ARI=76.4%, GT Split=29.2%, Pred Split=13.6%.
>
> **Q6: Related works.**
>
> A6: Thanks for the suggestion. We have cited and discussed this paper. Specifically, Yang et al. have proposed to learn visual concepts through visual dialog. In contrast, this paper learns visual concepts through question-answering pairs or captions. Most importantly, our model focuses on how concept learning and object discovery/representation learning bootstrap each other.
>
> ```
> [1] Justin Johnson, Bharath Hariharan, Laurens van der Maaten, Li Fei-Fei, C Lawrence Zitnick, and Ross Girshick. CLEVR: A diagnostic dataset for compositional language and elementary visual reasoning. In CVPR, 2017.
> [2] Jiayuan Mao, Chuang Gan, Pushmeet Kohli, Joshua B. Tenenbaum, and Jiajun Wu. The Neuro-Symbolic Concept Learner: Interpreting Scenes, Words, and Sentences from Natural Supervision. In ICLR, 2019
> [3] Kexin Yi, Jiajun Wu, Chuang Gan, Antonio Torralba, Pushmeet Kohli, and Josh Tenenbaum. Neural-Symbolic VQA: Disentangling reasoning from vision and language understanding. In NeurIPS, 2018
> [4] Sebastian Schuster, Ranjay Krishna, Angel Chang, Li Fei-Fei, and Christopher D. Manning. Generating Semantically Precise Scene Graphs from Textual Descriptions for Improved Image Retrieval. In Proceedings of the Fourth Workshop on Vision and Language (VL15), 2015.
> ```

---

### Author Response · Authors · 2020-11-25
**General Response**

We thank all reviewers for their comments. Prior to a detailed response to individual questions, we first clarify the goal of our paper and address some common concerns raised by the reviewers.

Previous methods on unsupervised object-centric representation learning have been mostly focusing on learning representations from a collection of images composed of similar objects. However, these methods are sensitive to hyperparameter choices and often fail when objects have complex geometries or translucent parts. As a result, these models tend to split an object or merge multiple objects. The visualization in Figure 1 on https://sites.google.com/view/lorl-iclr/ shows the result of the best-performing unsupervised algorithm (Slot Attention) on the challenging Shop-VRB-Simple dataset. Our quantitative evaluation metrics also suggest this.

Inspired by the joint object individuation and concept learning in human infant development [1], in this paper, we have proposed to mediate the object-centric representation learned with unsupervised signals, such as image reconstruction. Our systematic study has illustrated how language can improve model performance on unsupervised object discovery/segmentation, in challenging situations where objects have complex geometries and have multiple parts. Compared with other object-centric language learning works [2], LORL is able to ground linguistic units on objects without any prior knowledge or supervision on object detection/segmentation.

Q1: Types of Supervision (R3, R4).
A1: We agree with the reviewers that LORL received additional supervision from language, compared with baselines for unsupervised object discovery. However, our goal is not to improve existing methods, but to demonstrate how concept learning from language and unsupervised object discovery can bootstrap each other. Both of our quantitative numbers and qualitative visualizations support our claims. Furthermore, we have compared with two additional baselines on the visual reasoning task in our revision (see Q3 below). LORL outperforms the baseline that uses the same amount of supervision signals and archives a comparable result with baselines trained with groundtruth object segmentation annotations.

Q2: Ablation study on the objectness score (R3, R4).
A2: Per request, we have included an additional ablation study on objectness scores. Specifically, we compare two models: the original Slot Attention model and the Slot Attention model augmented with the proposed objectness score module. Both models are trained using images only (there is no language in the loop), on the Shop-VRB-Simple dataset. The following table summarizes the result. The objectness module alone does not contribute to the segmentation performance.


|                |   ARI  | GT split | Pred Split |
|:--------------:|:------:|:--------:|:----------:|
|       SA       | 83.51% |  15.68%  |   13.19%   |
| SA + Obj score | 83.40% |  16.57%  |   12.71%   |

On the other hand, although LORL without the objectness score achieves a comparable performance in terms of QA accuracy, the objectness score is crucial if we want to evaluate how models discover objects in images. Additionally, we show that, on the Shop-VRB-Simple dataset, LORL+SA shows significant improvement in concept quantization.

Concretely, we extract a scene graph for each scene, where each node corresponds to a detected object. We represent each node as a collection of concepts associated with the object (e.g., large brown wooden chopping board). We heuristically remove nodes that are not associated with any concepts (by treating them as “background” objects). We then compute the precision and the recall of the reconstructed scene graph by comparing it with the groundtruth one. We say a detected object node matches a groundtruth object node if their associated concept sets have an IoU that is greater than 0.75. Please refer to Appendix D for more details.

LORL+SA without the objectness score module has much more false positive detections that are backgrounds or parts of objects.

|                  | Precision@0.75 | Recall@0.75 |
|:----------------:|:--------------:|:-----------:|
| LORL + SA (No Obj. Score) |     54.13%     |    86.4%    |
|      LORL + SA     |     79.37%     |    84.02%   |


Q3: More baselines on visual reasoning (R3 R4).
A3: Thanks for the great suggestion. We have included two new baselines IEP [2] and NS-CL [3] for reference on the Shop-VRB-Simple dataset. All models are trained with the same set of question-answer pairs. Note that NSCL has the access to a pretrained object detection module, while LORL and IEP do not. LORL outperforms IEP, which is trained with exactly the same amount of supervision as ours. It also achieves a comparable result as NS-CL.

LORL + SA: 92.72%
IEP: 78.3%
NS-CL: 97.9%

---

> ### Author Response · Authors · 2020-11-25
> **General Response (Cont')**
>
> ```
> [1] Fei Xu. Sortal Concepts, Object Individuation, and Language. Trends in Cognitive Sciences 11.9 (2007): 400-406.
> [2] Justin Johnson, Bharath Hariharan, Laurens van der Maaten, Li Fei-Fei, C Lawrence Zitnick, and Ross Girshick. CLEVR: A Diagnostic Dataset for Compositional Language and Elementary Visual Reasoning. In CVPR, 2017.
> [3] Jiayuan Mao, Chuang Gan, Pushmeet Kohli, Joshua B. Tenenbaum, and Jiajun Wu. The Neuro-Symbolic Concept Learner: Interpreting Scenes, Words, and Sentences from Natural Supervision. In ICLR, 2019.
> ```

---

### Author Response · Authors · 2020-11-25
**Revision Note**

We thank all reviewers for their thoughtful reviews and constructive suggestions. We have revised our manuscript to include the following changes:

- Replace "Framework" with "Learning Paradigm".
- Minor typo fixes and sentence rephrasing.
- Detailed description of the training objectives on the PartNet-Chairs dataset. (Appendix C)
- Detailed description of the proposed GT/Pred split ratio metrics. (Appendix D, section “GT and Pred split ratios”)
- Detailed description of the referring expression comprehension task. (Appendix D, section “Referring expression comprehension”)
- Additional random seeds for the PartNet-Chairs dataset. (Table 2)
- Evaluation of the semantic parsing module. (Section 5.1)
- Ablation study on the second training phase. (Appendix B)
- Ablation study on objectness score. (Appendix E, section “Objectness score”, , Table 8)
- Ablation study on different question types and the number of question-answer pairs. (Appendix E, section “Question type”, Table 9)
- Additional results for visual reasoning baselines. (Appendix E, section “visual reasoning baselines”, Table 11)
- Additional results for integration with other unsupervised object-centric representation learning models. (Appendix E, section “Integrating with SPACE”, Table 12)
- Additional results for other ways of using language supervisions. (Appendix E, section “Baseline using language supervision”)
- Additional results for concept quantification accuracy. (Appendix E, section “concept quantification”, Table 13)

---

### Decision · Program_Chairs · 2021-01-07
**Final Decision**

**Decision:**

Reject

**Comment:**

While the paper's topic is on a topic of interest and presents an evaluation on three synthetic datasets,  PartNet-Chairs, Shop-VRB-Simple, CLEVR dataset, several concerns and weaknesses remain after the author response.

Main Concern and Weaknesses:
* The main improvement comes from the additional supervision provided by language, which provides a strong supervision signal as the language is scripted and the parser has nearly "perfect accuracy (>99.9%) on test questions/captions", as the authors state.
* Limited contribution: combination of MONet/Slot-Attention with NS-CL;
* Experiments limited to synthetic images with no background (relatively easy to segment) and synthetic (templated) language (easy to parse). This is especially concerning when the task is segmentation and the supervision comes from templated language, making it a strong supervision signal.
* The positive impact of the objectness score on performance was not sufficiently demonstrated
* Additionally, in the final discussion phase, reviewers were concerned that the with limited visual reasoning training on a subset of 25% of the data, reduces the performance [I note that I did not take this as the decisive reason for rejection as the authors did not have a chance to respond to this concern but the authors should discuss this in any future version of the paper]

The paper initially received borderline and reject scores and the authors took significant effort to address several of the comments of reviewers. While the paper was improved several of the main concerns remained and reviewers recommended reject after reading the author response and each others comments.

I agree to the concerns and recommend reject.